# Vti1a/b regulate synaptic vesicle and dense core vesicle secretion via protein sorting at the Golgi

Javier Emperador-Melero[1], Vincent Huson[2], Jan van Weering[2], Christian Bollmann[3], Gabriele Fischer von Mollard[3], Ruud F. Toonen [1] & Matthijs Verhage [1,2]

The SNAREs Vti1a/1b are implicated in regulated secretion, but their role relative to canonical exocytic SNAREs remains elusive. Here, we show that synaptic vesicle and dense-core vesicle (DCV) secretion is indeed severely impaired in Vti1a/b-deficient neurons. The synaptic levels of proteins that mediate secretion were reduced, down to 50% for the exocytic SNARE SNAP25. The delivery of SNAP25 and DCV-cargo into axons was decreased and these molecules accumulated in the Golgi. These defects were rescued by either Vti1a or Vti1b expression. Distended Golgi cisternae and clear vacuoles were observed in Vti1a/b-deficient neurons. The normal non-homogeneous distribution of DCV-cargo inside the Golgi was lost. Cargo trafficking out of, but not into the Golgi, was impaired. Finally, retrograde Cholera Toxin trafficking, but not Sortilin/Sorcs1 distribution, was compromised. We conclude that Vti1a/b support regulated secretion by sorting secretory cargo and synaptic secretion machinery components at the Golgi.

[1] Departments of Functional Genomics, Clinical Genetics, VUmc, Center for Neurogenomics and Cognitive Research, Amsterdam Neuroscience, VU University Amsterdam and VU Medical Center, de Boelelaan 1087, 1081 HV Amsterdam, The Netherlands. [2] Clinical Genetics, VUmc, Center for Neurogenomics and Cognitive Research, Amsterdam Neuroscience, VU University Amsterdam and VU Medical Center, de Boelelaan 1087, 1081 HV Amsterdam, The Netherlands. [3] Department of Biochemistry III, Bielefeld University, 33615 Bielefeld, Germany. Correspondence and requests for materials should be addressed to M.V. (email: matthijs@cncr.vu.nl)

Regulated secretion of signaling molecules from synaptic vesicles (SVs) and dense-core vesicles (DCVs) is the primary means of neuronal communication. SVs store neurotransmitters and are locally recycled after exocytosis[1–3], whereas DCVs, which contain neuropeptides and neurotrophins[4,5], are continuously generated at the trans-Golgi[6,7]. Immature DCVs are proposed to undergo initial fusion reactions, homotypic fusion[8,9] or heterotypic fusion to endosomes[10], before they are fusion-competent. Fusion requires formation of SNARE complexes consisting of four SNARE protein domains (R, Qa, Qb, and Qc[11]). In addition to the canonical exocytic SNARE proteins (VAMP2, Syntaxin-1, and SNAP25[1,3]), other R- and Q-SNAREs are implicated in regulated secretion. However, how these additional SNAREs regulate secretion in relation to the canonical exocytic SNAREs is still poorly understood.

In this context, the Qb-SNARE Vti (VPS10 interacting) proteins are particularly enigmatic. Two Vti proteins are ubiquitously expressed in mammals, Vti1a and Vti1b[12–14]. These proteins function in endo-lysosomal trafficking[15–19] and combined gene inactivation in mice triggers neurodegeneration and perinatal death[12]. Vti1a knockdown results in reduced spontaneous SV fusion[20,39]. However, it is unclear which domain Vti1a would contribute to exocytic SNARE complexes because SNAP25 supplies the Qb-SNARE domain for spontaneous SV fusion[21–23]. Vti1a knock-out chromaffin show reduced exocytosis by unknown pathways that result in decreased numbers of secretory granules[24]. Hence, it is possible that, also in neurons, dysregulation of Vti1a/b-dependent upstream pathways contributes to downstream secretion defects.

Here, Vti1a/b-null neurons were used to test this hypothesis. We found that the number of DCVs and synapses, and the synaptic levels of proteins that mediate secretion were all reduced. The secretion efficiency of SVs and DCVs was severely impaired. The delivery of DCV-cargo and SNAP25 into axons was reduced. DCV-cargo and SNAP25 accumulated in the Golgi. These phenotypes were rescued by expression of either Vti1a or Vti1b. Furthermore, Golgi cisternae were distended and the export of cargo from the Golgi was impaired. We conclude that Vti1a/b regulate secretion by sorting secretory cargo and proteins of the secretion machinery out of the Golgi.

## Results

**Fewer synapses and DCVs in Vti1a/b-deficient neurons.** To study the role of Vti1a and Vti1b in regulated secretion, we used Vti1a/Vti1b double knock-out (DKO) mice[12], with double heterozygotes (DHZ) as controls (Supplementary Figure 1A). Expression of Vti1a or Vti1b in DKO neurons at 1 day in vitro (DIV1) using lentiviral vectors produced perinuclear expression, similar to endogenous proteins (Supplementary Figure 1B). Cell death was observed in DKO neurons between DIV3-10 (Supplementary Figure 1C), affecting glutamatergic and GABAergic subpopulations equally (Supplementary Figure 1D-E). The surviving DKO neurons had 30% shorter dendrites at DIV-14 (Fig. 1a; Supplementary Figure 1F-G, 2A), without altered arborization (Supplementary Figure 2B). This reduction was rescued by either Vti1a or Vti1b expression (Fig. 1a; Supplementary Figure 1F-G, 2A). The axonal length was reduced to a similar extent and was only partially rescued (Supplementary Figure 1F, H, I). The length reduction was not a property of the surviving neurons, as DIV-4 neurons had similar alterations (Supplementary Figure 1J–M).

Synaptic vesicles (SVs) were quantified by staining for endogenous Synaptophysin-1 (Fig. 1a), and DCVs by expressing an established DCV cargo reporter, Neuropeptide-Y (NPY)

fused to pHluorin (Fig. 1f), which co-localizes with endogenous DCV markers[21,25,26]. At DIV-14, expression of both markers was punctate (Fig. 1a, f). In DKO neurons, the number of Synaptophysin-1 and NPY-pHluorin puncta was reduced by 65% and 50%, respectively (Supplementary Figure 2C, D), which resulted in 40% (Fig. 1b) and 15% (Fig. 1g) reductions after compensating for differences in neurite length. Puncta distribution relative to the soma was unaltered (Fig. 1e, j). Furthermore, puncta intensity in DKO neurons was 30% lower for Synaptophysin-1 (Fig. 1c, d) and 50% lower for NPY-pHluorin (Fig. 1h, i). The endogenous DCV cargo BDNF was analyzed to corroborate the differences observed using the DCV reporter. The proportion of BDNF-positive neurons was similar between groups ($83.2 \pm 7.1$, $79.7 \pm 9.3$, $71.7 \pm 2.5$, $81.1 \pm 12.8\%$ for DHZ, DKO and rescue with Vti1a or Vti1b, respectively). The density and intensity of BDNF puncta were reduced in DKO neurons (Supplementary Figure 2E–I), without alterations in distribution (Supplementary Figure 2K). Expression of Vti1a or Vti1b rescued all these parameters (Fig. 1; Supplementary Figure 2A–J), except for DCV puncta intensity, which was only partially rescued by Vti1b (Fig. 1h–i; Supplementary Figure S2 H, I).

At the ultrastructure level, random cross sections of synapses contained similar numbers of SVs, docked SVs per synapse, active zone length and postsynaptic density length between DHZ and DKO (Fig. 1k–m, Supplementary Figure 2K, L). However, synaptic profiles contained DCVs twofold less often in DKO neurons (DHZ, $30.27 \pm 2.10\%$; DKO, $16.05 \pm 5.81\%$) and, among DCV-containing micrographs, DKO synapses contained 40% fewer DCVs (Fig. 1n). The DCV diameter was not altered in DKO neurons (Fig. 1O, P).

These results show that Vti1a/b deficiency results in in vitro cell death, and that the surviving neurons are smaller and have fewer synapses and DCVs. Furthermore, expression of either Vti1a or Vti1b rescues virtually all these phenotypes equally well.

**Reduced SV and DCV secretion in Vti1a/b-deficient neurons.** We investigated how regulated secretion is affected by loss of Vti1a/b expression by quantifying SV and DCV fusion. To study SV fusion, we applied live dual-color imaging of Synaptophysin-pHluorin[27] (SypHy) and FM4-64-labeled SVs (Supplementary Figure 3A), and whole-cell patch clamp. SypHy-fluorescence was quenched under resting conditions (Fig. 2a; Supplementary Movie 1). High frequency stimulation (HFS, 100 action potentials (APs) at 40 Hz) triggered SV fusion from most synapses (Fig. 2a). A higher percentage of non-responsive synapses was observed in DKO neurons (Fig. 2b). In responsive synapses, the fraction of SVs that fused was 50% lower (Fig. 2c, d). Neither kinetics of fluorescence increase nor fluorescence decay after HFS, a measure for SV endocytosis and re-acidification[27], were affected in DKO neurons (Fig. 2e; Supplementary Figure 3E). Consistent with SypHy, DKO neurons also showed reduced FM4-64 uptake upon 60 mM $K^+$ depolarization, and reduced release upon HFS (Supplementary Figure 3F-I). These parameters were rescued by expression of either Vti1a or Vti1b (Fig. 2a–d; Supplementary Figure 3B-I).

In whole-cell patch clamp recordings, evoked transmitter release upon single AP or HFS (100APs, 40 Hz) stimulation was decreased by 80–90% (Fig. 2f, g; Supplementary Figure 3J, L, M). The amplitude and charge transfer of spontaneous transmitter release was 20–25% lower (Fig. 2f, h; Supplementary Figure 3O). After correcting for the decreased SV charge, evoked release remained reduced to a similar extent, 80–90% (Supplementary Figure 3K, N). The frequency of spontaneous fusion events was

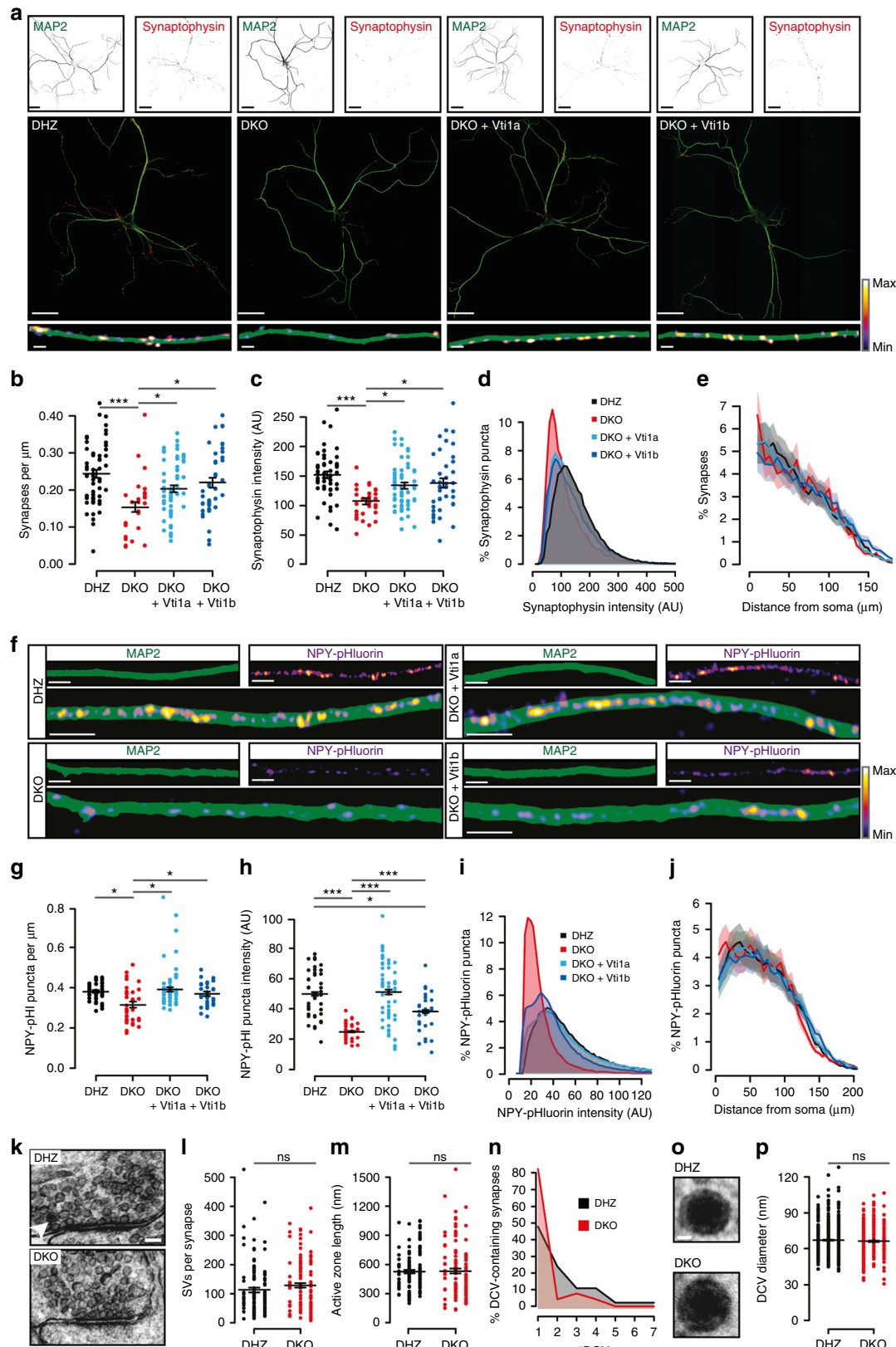

reduced by 65% (Fig. 2f, i). The readily releasable pool (RRP), quantified as charge transferred during hypertonic sucrose superfusion, was almost seven fold smaller (Fig. 2j; Supplementary Figure 3P). The release probability, calculated as charge transfer evoked by one AP over the RRP, was 45% lower (Fig. 2k).

Finally, the paired pulse ratio was larger in DKO neurons (Supplementary Figure 3L, Q).

To investigate DCV secretion, we performed live imaging of NPY-pHluorin-labeled DCVs. NPY-pHluorin fluorescence was initially quenched (Fig. 3a; Supplementary movie 2). HFS

**Fig. 1** Vti1a/b-deficient neurons have shorter neurites and contain less synapses and DCVs. **a** Representative examples of single DIV-14 neurons immunostained for MAP2 and Synaptophysin-1. **b** Lower synapse density in DKO neurons (DHZ: 0.24 ± 0.01, $n = 57$; DKO: 0.15 ± 0.01, $n = 32$; DKO + Vti1a: 0.20 ± 0.01, $n = 60$; DKO + Vti1b: 0.22 ± 0.01 synapses/μm dendrite, $n = 43$; Kruskal–Wallis). **c** Reduced Synaptophysin-1 puncta intensity in DKO neurons (DHZ: 151.71 ± 8.82, $n = 57$; DKO: 106.96 ± 5.16, $n = 32$; DKO + Vti1a: 133.47 ± 5.24, $n = 60$; DKO + Vti1b: 137.75 ± 8.09 AU, $n = 43$; ANOVA). **d** Histogram showing higher proportion of low-intensity Synaptophysin-1 puncta in DKO neurons. **e** Sholl analysis showing similar synapse distribution, with a non-significant increase at 5 μm in DKO (6.40 ± 0.46) compared to DHZ (5.01 ± 0.33), DKO + Vti1a (5.29 ± 0.34) and DKO + Vti1a (4.76 ± 0.29%; Kruskal–Wallis). **f** Representative stretches of single DIV-14 neurons expressing NPY-pHluorin and immunostained for MAP2. **g** Reduced NPY-pHluorin puncta density in DKO neurons (DHZ: 0.38 ± 0.01, $n = 42$; DKO: 0.32 ± 0.02, $n = 37$, DKO + Vti1a: 0.39 ± 0.01, $n = 59$; DKO + Vti1b: 0.37 ± 0.01 puncta/μm neurite, $n = 34$; Kruskal–Wallis). **h** Reduced NPY-pHluorin puncta intensity in DKO neurons (DHZ: 49.8 ± 12.0, $n = 42$; DKO: 24.9 ± 4.4, $n = 37$, DKO + Vti1a: 51.0 ± 15.7, $n = 59$; DKO + Vti1b: 38.3 ± 9.7 AU, $n = 34$; Kruskal–Wallis). **i** Histogram showing reduced NPY-pHluorin puncta intensity in DKO neurons. **j** Sholl analysis showing similar distribution of NPY-pHluorin puncta between groups. **k** Representative synaptic micrographs of single DIV-14 neurons. Arrowhead points at DCV. **l** Similar amount of SVs per synapse between DHZ (113.53 ± 7.71, $n = 122$) and DKO neurons (129.28 ± 8.11, $n = 119$; Mann–Whitney). **m** Similar active zone length between DHZ (527.17 ± 18.67, $n = 122$) and DKO neurons (534.20 ± 25.26 nm, $n = 119$; Mann–Whitney). **n** Histogram of DCVs per synapse in DCV-containing synapses showing more DCVs in DHZ (2.11 ± 0.22) than DKO neurons (1.35 ± 0.17). **o** Representative examples of DCVs. **p** Similar DCV diameter between DHZ (67.11 ± 0.48, $n = 626$) and DKO neurons (66.17 ± 0.50 nm, $n = 502$; Mann–Whitney). Bars show mean ± SEM. Scatterplots and columns represent individual neurons and litters, respectively.*$p < 0.05$; **$p < 0.01$; ***$p < 0.001$. Scale bar = 40 μm (**a**, overview), 2 μm (**a**, zoom), 5 μm (**h**), 100 nm (**k**), 20 nm (**o**). Intensity color-code used for Synaptophysin-1 zooms and NPY-pHluorin

stimulation (16 trains of 50APs at 50 Hz), reported to elicit efficient DCV fusion[21,25,26], triggered similar calcium transients in all groups (Supplementary Figure 4A-F). Fusion of individual DCVs, characterized by dequenching of NPY-pHluorin puncta (Supplementary Figure 4G), was 85% lower in DKO neurons (Fig. 3a, b). The DCV fusion probability, calculated as DCV fusion events (Fig. 3b) over total DCV pool (Fig. 3c), was 3.5-fold lower (Fig. 3d; Supplementary movie 2). The release kinetics were similar between groups, with ~30% events occurring during the first 3 stimulation trains (Fig. 3e, f). Individual fusion events were more transient in DKO neurons (Supplementary Figure 4H). However, the event amplitude was unaffected (Supplementary Figure 4, I, J). The fraction of fusion events that occurred at synapses (within the point spread function of the synapse marker Synapsin-ECFP, Supplementary Figure 4K), was 30% in DKO neurons and 50% in DHZ (Supplementary Figure 5L). All these phenotypes were rescued by expression of either Vti1a or Vti1b (Fig. 3a–f; Supplementary Figure 6H-L), except for the total DCV pool, which was only partially rescued by Vti1b. A reduction in DCV fusion was not observed in Vti1a or Vti1b single KO neurons (Supplementary Figure 4M-R).

These results indicate that DKO neurons have impaired secretion, characterized by smaller readily releasable SV pools, decreased SV release probability, reduced spontaneous SV fusion frequency and amplitude, and decreased DCV fusion probability. Vti1a/b are dispensable for fusion kinetics during stimulation, SV endocytosis and acidification or DCV cargo loading and acidification.

**Lower levels of exocytic proteins in Vti1a/b DKO neurons.** To understand why regulated secretion is impaired in DKO neurons, we measured the levels of eight synaptic proteins involved in exocytosis by immunocytochemistry, using Synaptophysin-1 as synaptic marker. DKO neurons had a significant reduction in staining intensity for synaptic SNAP25 (61.3%), Munc13-1 (46.1%), Bassoon (35.5%), Synaptotagmin-1 (29.8%), and RIM1/2 (19.2%) (Fig. 4; Supplemental Figure A–H & Table1). These reductions were rescued by either Vti1a or Vti1b expression. The staining intensity of extrasynaptic SNAP25 was decreased to a similar extent (Supplementary Figure 5I). Staining for Synaptobrevin-2, Syntaxin-1, and Munc18-1 showed a trend towards reduced intensity (19.3, 16.4, and 10.4%, respectively, not significant). The reduction in synaptic protein levels correlated with known protein half-lifes[28] ($r^2 = 0.719$, $p < 0.0001$; Supplementary Figure 5J).

To test whether the reduced secretion and synaptic protein levels were features specific of the surviving DKO neurons, Vti1a and Vti1b were expressed in DKO neurons after most cell death had occurred (DIV10; Supplementary Figure 6A). This late expression rescued the fraction of responsive synapses, SV fusion in responsive synapses (Supplementary Figure 6B–D) and levels of SNAP25 and Munc13-1 (Supplementary Figure 6E–H).

These results indicate that Vti1a/b are required to maintain normal levels of secretion machinery components, and that the reduced number of responsive synapses, synaptic secretion and synaptic protein levels are not features specific for surviving DKO neurons.

**Reduced protein influx into axons in Vti1a/b DKO neurons.** Increased protein degradation or reduced delivery to synapses are the two main explanations for the decreased synaptic levels of proteins that mediate secretion. To address the first possibility, we blocked protein translation for 24 h with cycloheximide. Cycloheximide treatment reduced synaptic SNAP25 and Munc13-1 levels to a similar extent in DHZ and DKO neurons (Supplementary Figure 5K-O). To address protein delivery, SNAP25-EGFP transport was monitored in DIV-5 neurons after photobleaching the proximal axon, identified with the axon initial segment marker NaV$_{II/III}$-mCherry (Fig. 5a). SNAP25 was selected as reporter because its levels in DKO neurons were decreased the most. SNAP25-EGFP puncta were detected entering the axon in all groups, at an average speed of around 1 μm/s (Fig. 5b). In DHZ neurons, 3.9 puncta/min entered the axon, as opposed to 2.2 in DKO neurons (Fig. 5c).

The reduced number of DCVs detected in DKO neurons (Fig. 1) may be the consequence of a similar inefficient delivery to axons. To address this, we expressed the DCV-reporter NPY-mCherry and NaV$_{II/III}$-YFP (Fig. 5d). NPY puncta entered the axon in all groups, at a velocity of 1.0–1.3 μm/s (Fig. 5e). However, similar to SNAP25-EGFP, 40% fewer NPY puncta entered the axon in DKO neurons (Fig. 5f). NPY delivery to the axon was rescued by expression of either Vti1a or Vti1b (Fig. 5d, f). These results indicate that Vti1a/b control the efficient delivery of SNAP25 and DCV cargo to axons.

**Abnormal Golgi and cargo accumulation in Vti1a/b DKO neurons.** To understand the reduced cargo delivery to axons in DKO neurons, we analyzed the location of Vti proteins. 90% of the Golgi-marker MannosidaseII-ECFP (ManII-ECFP) co-localized with Vti1a and Vti1b (Fig. 6a, b; Supplementary

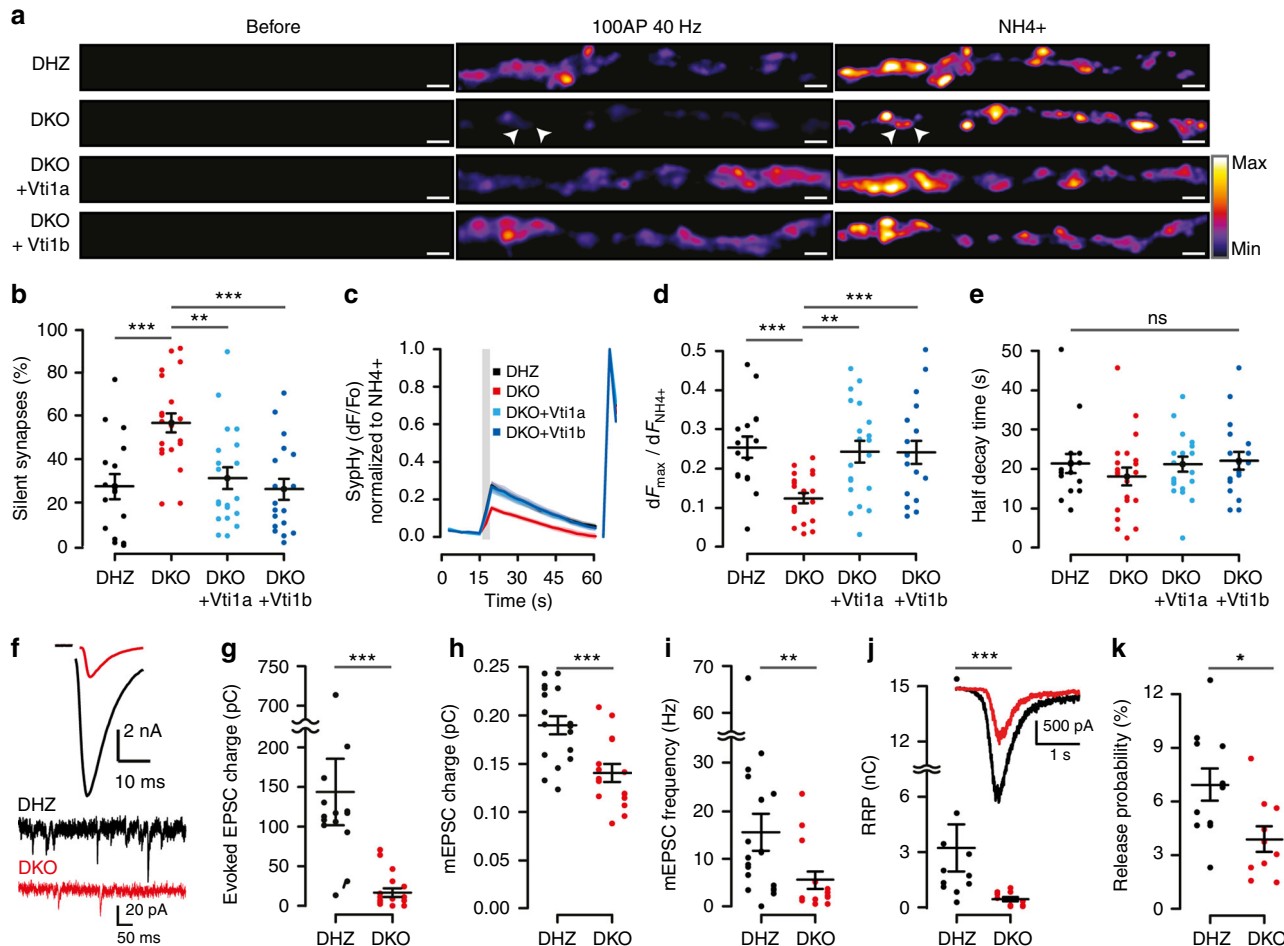

**Fig. 2** SV fusion is impaired in Vti1a/b-deficient neurons. **a** Examples of SypHy-labeled synapses from single DIV-14 neurons before, during HFS (100AP, 40 Hz) and $NH_4^+$ superfusion. Arrowheads point at examples of silent synapses. Intensity color-code used for SypHy. Scale bar = 2 μm. **b** Increased percentage of silent synapses in DKO neurons (DHZ: 56.65 ± 4.45, n = 17; DKO: 27.28 ± 5.77, n = 21; DKO + Vti1a: 31.21 ± 5.14, n = 20; DKO + Vti1b: 26.03 ± 4.95%, n = 19; ANOVA). **c** Average SypHy traces of SV fusion in responsive synapses upon HFS (gray rectangle) normalized to the maximum intensity upon $NH_4^+$ superfusion (right traces). Shaded regions indicate SEM. **d** Reduced fraction of SV fusion in responsive synapses, measured as maximum HFS intensity over $NH_4^+$ response, in DKO neurons (DHZ: 0.25 ± 0.03, n = 17; DKO: 0.12 ± 0.01, n = 21; DKO + Vti1a: 0.24 ± 0.03, n = 20; DKO + Vti1b: 0.24 ± 0.03, n = 19; ANOVA). **e** Similar half decay time of the maximum HFS response between groups (DHZ: 21.45 ± 0.44, n = 17; DKO: 18.06 ± 2.25, n = 21; DKO + Vti1a: 21.24 ± 1.20, n = 20; DKO + Vti1b: 22.13 ± 2.23 s, n = 19; Kruskal–Wallis). **f** Representative traces of evoked (top) and spontaneous mini (bottom) excitatory postsynaptic currents (EPSC). **g** Reduced evoked EPSC charge transfer in DKO neurons (DHZ: 145.03 ± 42.66, n = 15; DKO: 18.36 ± 5.19 pC; Mann–Whitney). **h** Reduced spontaneous mini EPSC (mEPSC) charge transfer in DKO neurons (DHZ: 0.19 ± 0.01, n = 17; DKO: 0.14 ± 0.01 pC; t-test). **i** Reduced mEPSC frequency in DKO neurons (DHZ: 15.45 ± 3.84, n = 18; DKO: 5.55 ± 1.79 Hz, n = 15; Mann–Whitney). **j** The readily releasable pool (RRP), calculated as total sucrose-induced (500 mM) charge transfer, is reduced in DKO (DHZ: 3.23 ± 1.28, n = 11; DKO: 0.49 ± 0.12 nC, n = 10; Mann–Whitney). Inset shows representative examples. **k** SV release probability, calculated as charge transfer during the first evoked over RRP, is reduced in DKO neurons (DHZ: 6.9 ± 0.9, n = 11; DKO: 3.9 ± 0. 7%, n = 10; t-test). Bars show mean ± SEM. Scatterplots and columns represent individual neurons and litters, respectively. *p < 0.05; **p < 0.01; ***p < 0.001

Figure 7A). Vti1a and Vti1b strongly co-localized at ManII-ECFP positive regions (93%), while in the rest of the neuron co-localization was low (22%, Fig. 6c). In ManII-ECFP negative regions, Vti1a and Vti1b puncta partially co-localized with endo-lysosomal markers (Supplementary Figure 7), as expected[15–19].

Because of the preferential localization of Vti1a/b to the Golgi, we examined the structure of this organelle in DKO neurons. At DIV-14, the total areas stained for the cis-Golgi marker GM130 and the trans-Golgi marker TGN46 were reduced in DKO neurons (Fig. 6d–f), but the staining intensity of these markers was increased (Supplementary Figure 8A, B). Expression of either Vti1a or Vti1b rescued these differences (Fig. 6d–f; Supplementary Figure 8A, B). A smaller GM130-stained area was also observed in acute hippocampal slices of DKO neurons (Supplementary Figure 8C). At the ultrastructural level, distended Golgi

cisternae were frequently observed in DKO neurons (Fig. 6G, H), with clear vacuoles in the proximity of the Golgi (Fig. 6g, i). Occasionally, multilayer membrane structures were observed in DKO neurons (Supplementary Figure 8D). Mitochondria appeared normal (Fig. 6g, j). These data indicate that Vti1a/b localize largely to the Golgi, and that the Golgi morphology is abnormal in DKO neurons.

We subsequently investigated a potential relationship between Golgi defects and the reduced cargo delivery to axons at DIV-5. At DIV-5, the areas stained by GM130, ManII-EGFP, and TGN46 were already smaller in DKO neurons (Supplementary Table 1). Endogenous SNAP25 was found in ManII- EGFP- and TGN46-positive areas (Fig. 6k). SNAP25 staining in these areas was greater than in the rest of the soma (Fig. 6k). In DKO neurons, the ratio of SNAP25 expression in ManII- EGFP- or TGN46-positive areas

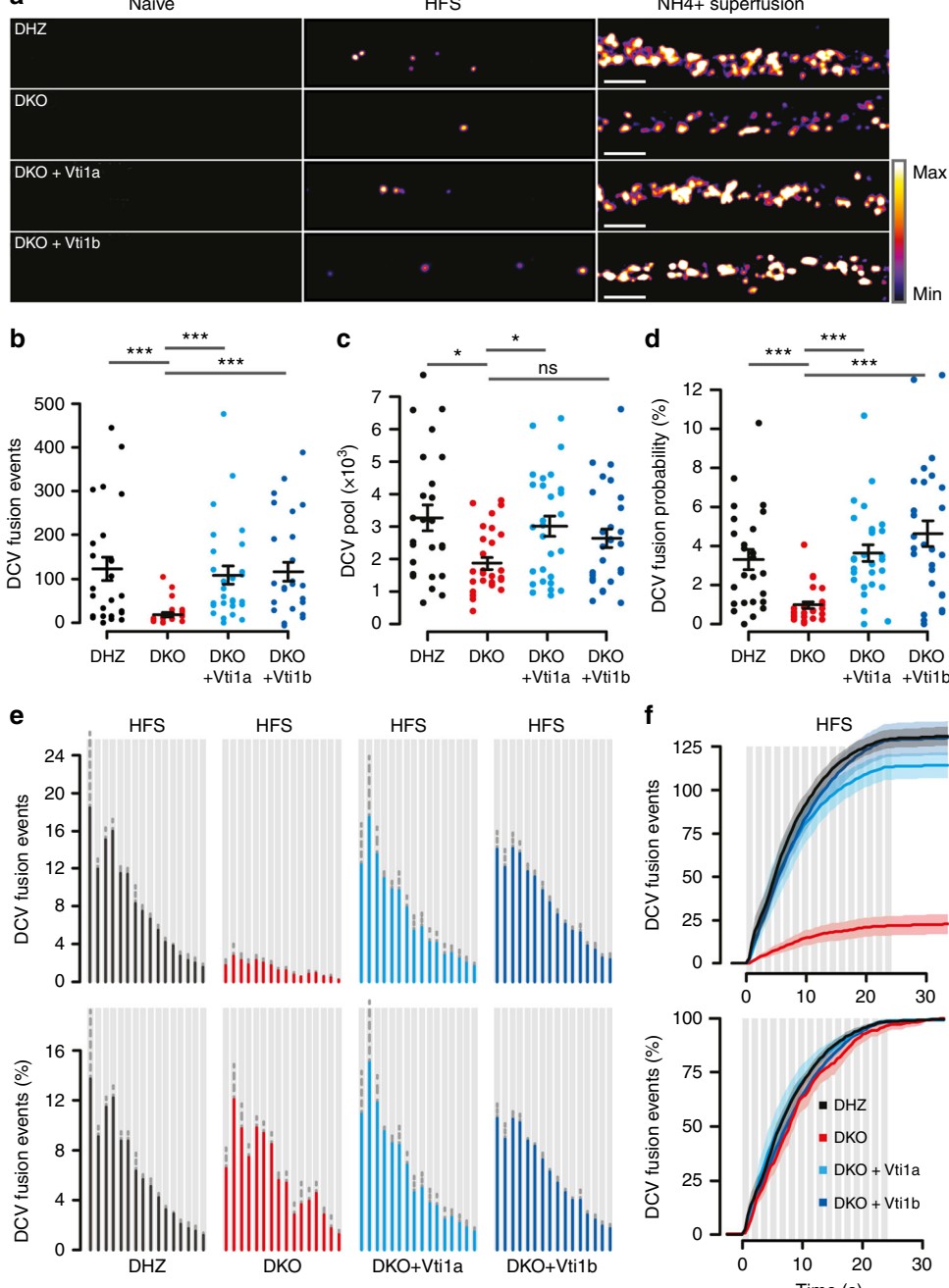

**Fig. 3** Reduced dense-core vesicle secretion in Vti1a/b-deficient neurons. **a** Representative examples of single DIV-14 neurons expressing NPY-pHluorin, before, during HFS (16 trans, 50APs at 50 Hz) and after $NH_4^+$ superfusion. Scale bar = 5 μm. Images show maximum projections of time-lapse recordings. **b** DCV fusion is reduced in DKO neurons (DHZ: 123.6 ± 26.4; DKO: 18.9 ± 4.5; DKO + Vti1a: 108.5 ± 20.7; DKO + Vti1b: 119.8 ± 22.1 fusion events/neuron; Kruskal–Wallis). **c** Reduced DCV pool, calculated upon $NH_4^+$ superfusion, in DKO neurons (DHZ: 3259.9 ± 395.8; DKO: 1865.0 ± 182.3; DKO + Vti1a: 2997.6 ± 314.7; DKO + Vti1b: 2697.0 ± 294.4 DCVs/neuron; Kruskal–Wallis). **d** Decreased DCV fusion probability in DKO neurons (DHZ: 3.29 ± 0.51; DKO: 0.96 ± 0.17; DKO + Vti1a: 3.60 ± 0.43; DKO + Vti1b: 4.59 ± 0.67%; Kruskal–Wallis). **e, f** Histogram and cumulative plots of the total (top) and fraction (bottom) of DCV fusion events per stimulation train (gray bars). Bars show mean ± SEM. Bullets and columns represent individual observations and independent litters, respectively. Gray bars represent 16 × 50 AP at 50 Hz stimulation. $N$ = 25, 29, 28, 27 for DHZ, DKO, DKO + Vti1a and DKO + Vti1b, respectively. *$p < 0.05$, ***$p < 0.001$

over the rest of the soma was 16% and 23% higher, respectively (Fig. 6l–m; Supplementary Figure 8E–G). NPY-pHluorin was also enriched in GM130- and TGN46-positive areas and the ratio of expression was greater in DKO than controls at GM130- (57%) and TGN46-positive areas (63%) (Fig. 6n–p, Supplementary Figure 8H–J). The expression ratios for SNAP25 and NPY were rescued by expression of either Vti1a or Vti1b (Fig. 6k–p). The

expression of the postsynaptic protein PSD95 and TrKB receptors were also decreased and, like NPY-pHluorin and SNAP25, the ratio of expression inside/outside of the Golgi was greater in DKO neurons (Supplementary Figure 9). Hence, in the absence of Vti1a/b, Golgi export appears to be compromised for both presynaptic and postsynaptic proteins, as well as DCV-cargo.

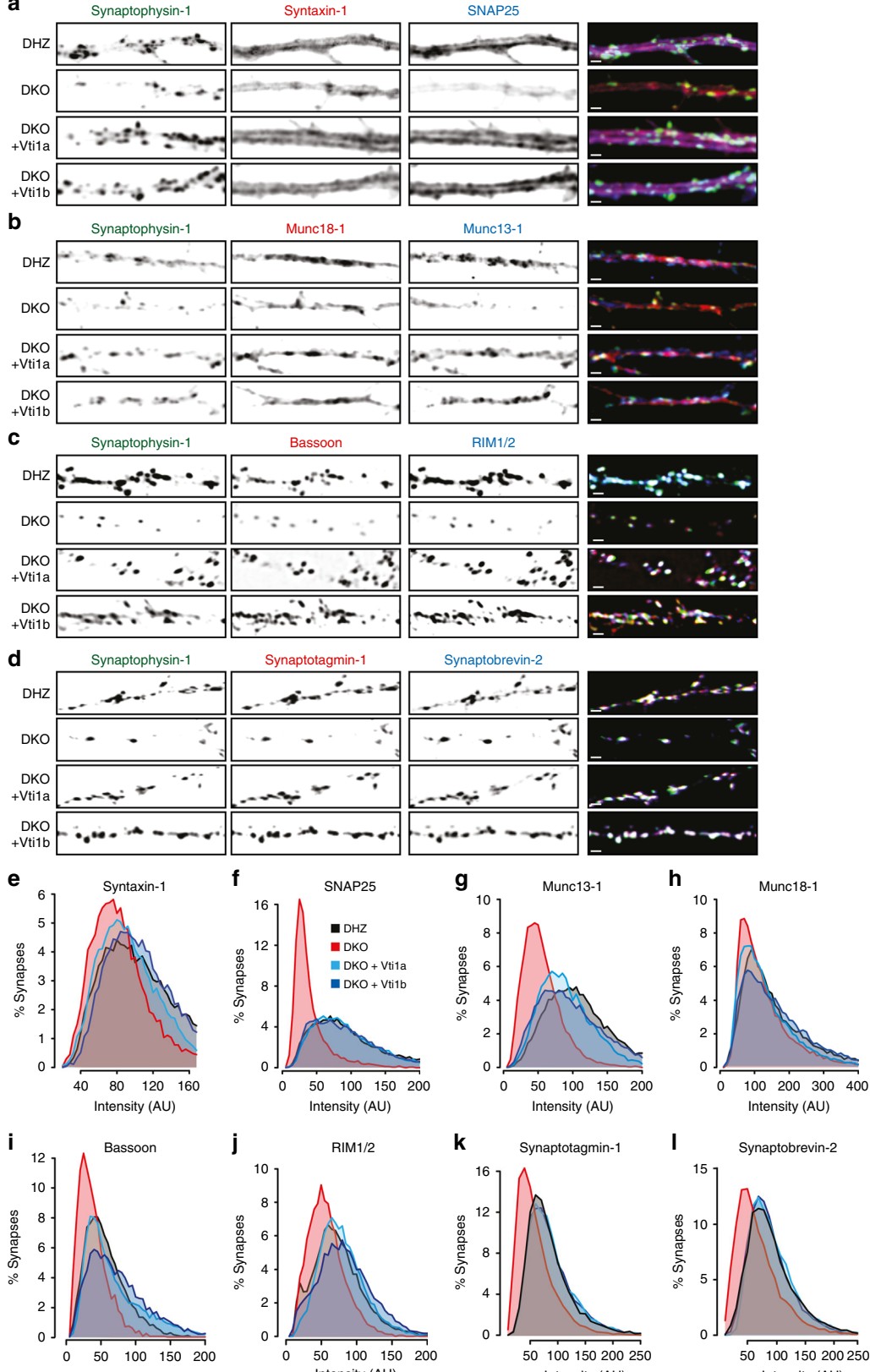

**Fig. 4** Reduced synaptic levels of proteins that mediate secretion in Vti1a/b-deficient neurons. **a–d** Representative examples of single DIV-14 neurons immunostained for Synaptophysin-1 as synaptic marker and Syntaxin-1 and SNAP25 (**a**), Munc18-1 and Munc13-1 (**b**), RIM1/2 and Bassoon (**c**), and Synaptobrevin-2 and Synaptophysin-1 (**d**). **e–l** Intensity distribution of the levels of Syntaxin-1 (**e**), SNAP25 (**f**), Munc13-1 (**g**), Munc18-1 (**h**), Bassoon (**i**), RIM1/2 (**j**), Synaptotagmin-1 (**k**), Synaptobrevin-2 (**l**) in all synapses (*n* = Over 15,000 synapses per protein and group). Statistical significance tested in Supplementary Figure 5. Scale bar = 2 μm

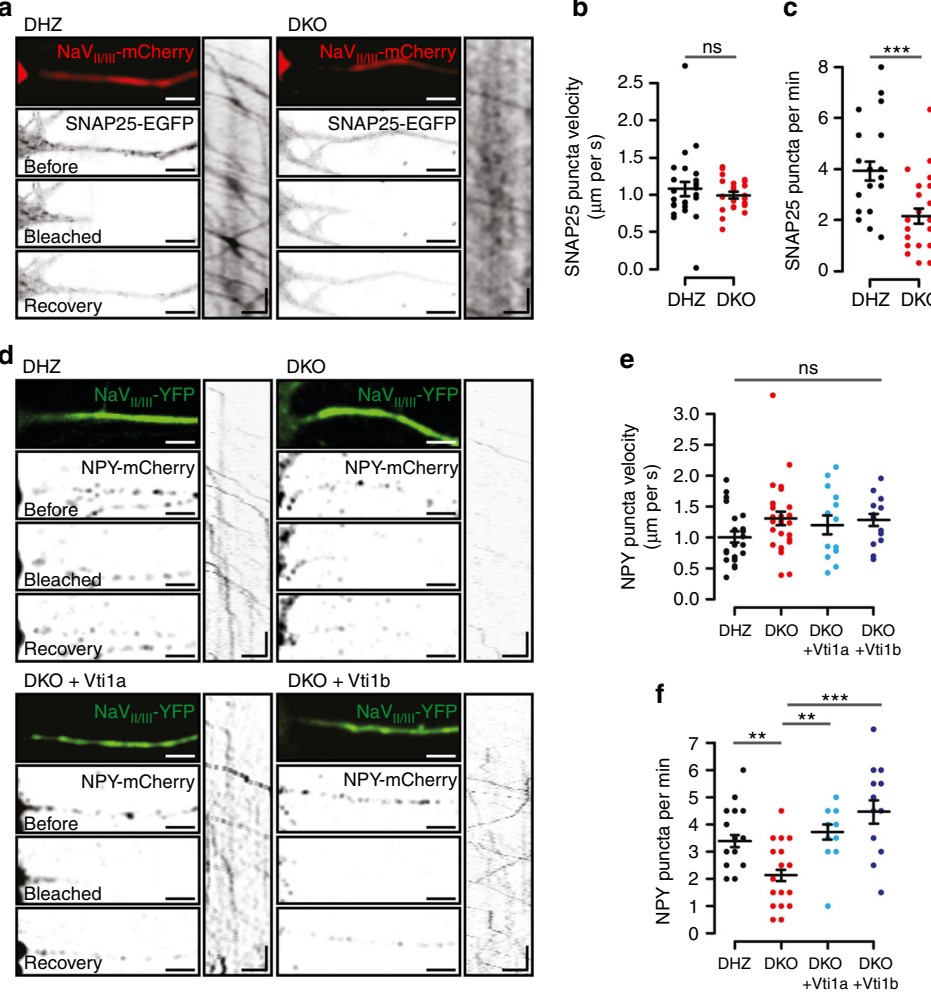

**Fig. 5** Reduced influx of SNAP25 and DCV cargo into axons in Vti1a/b-deficient neurons. **a** Representative examples of DIV-5 neurons, in mass cultures, expressing SNAP25-EGFP and the axonal-initial-segment (AIS) marker NaV$_{II/III}$-mCherry before, after photobleaching and three minutes after photobleaching. Kymographs show SNAP25-EGFP puncta in the AIS photobleached segment. **b** No differences in SNAP25 puncta velocity in the AIS (DHZ: 1.08 ± 0.10, $n = 24$; DKO: 0.99 ± 0.04 μm/s, $n = 24$; Mann–Whitney). **c** Less SNAP25 puncta enter the AIS in DKO neurons (DHZ: 3.93 ± 0.37, $n = 24$; DKO: 2.17 ± 0.30 puncta/min, $n = 24$; t-test). **d** Representative examples of DIV-5 neurons, in mass cultures, expressing the AIS marker NaV$_{II/III}$-YFP and NPY-mCherry before, after photobleaching and two minutes after photobleaching. Kymographs show NPY-mCherry puncta in the AIS photobleached segment. **e** No differences in NPY puncta velocity in the AIS (DHZ: 1.00 ± 0.09, $n = 23$; DKO: 1.30 ± 0.11, $n = 27$; DKO + Vti1a: 1.20 ± 0.15, $n = 14$; DKO + Vti1b: 1.28 ± 0.10 μm/s, $n = 15$; Wilcoxon). **f** Less NPY puncta enter the axon in DKO neurons (DHZ: 3.39 ± 0.23, $n = 23$; DKO: 2.13 ± 0.21, $n = 27$; DKO + Vti1a: 3.71 ± 0.28, $n = 14$; DKO + Vti1b: 4.47 ± 0.43 puncta/min, $n = 15$; Wilcoxon). Bars show mean ± SEM. Bullets and columns represent individual observations and independent litters, respectively. In **c** and **f**, some individual observations had the same value and overlapped in the graph. **$p < 0.01$, ***$p < 0.001$. Scale bar = 10 s and 5 μm

To further analyze this cargo accumulation, we acquired higher resolution images of the somata. The Golgi was labeled with GM130 and TGN46. Discrete NPY-pHluorin puncta were observed in somata, surrounding the Golgi area (Supplementary Figure 8K). The location of these somatic puncta relative to the Golgi was not altered in DKO neurons (Supplementary Figure 8L). Within the Golgi area, NPY-pHluorin fluorescence was non-homogeneous, with sub-areas of NPY-pHluorin enrichment (Supplementary Figure 8M). In DKO neurons NPY-pHluorin distribution was more homogeneous (Supplementary Figure 8M, N). NPY-pHluorin puncta co-localized to a similar (low) extent with the lysosomal marker Lamp1 (Supplementary Figure 8O-Q).

These data show that Vti1a/b are required for a normal Golgi structure and focal distribution of secretory cargo within sub-areas of the Golgi. In the absence of Vti1a/b, such cargo, and also SNAP25, accumulate at the Golgi.

**Reduced cargo export from the Golgi in Vti1a/b DKO neurons.** The accumulation and abnormal distribution of cargo at the Golgi in DKO neurons prompted us to investigate the trafficking to/from this organelle using the RUSH system[29]. NPY-EGFP fused to Streptavidin-binding protein (SBP) was co-expressed in DIV-5 neurons with the ER-resident protein KDEL fused to Streptavidin. The interaction between Strepta-vidin and SBP traps cargo at the ER[29] (Fig. 7a). Trafficking of the constitutive cargo, GPI, was studied in the same manner. Reversion of the SBP-Streptavidin binding by Biotin application triggered NPY or GPI transport to neurites (Fig. 7b, c). Prior to biotin application, the distribution of GPI and NPY was dispersed at the somata (Fig. 7d, i), consistent with ER retention[29]. Biotin application triggered GPI and NPY translocation to ManII-ECFP-labeled areas (Golgi) within minutes (Fig. 7d, i), with similar kinetics between groups (Fig. 7d–f, i–k). However, the NPY export kinetics were slower

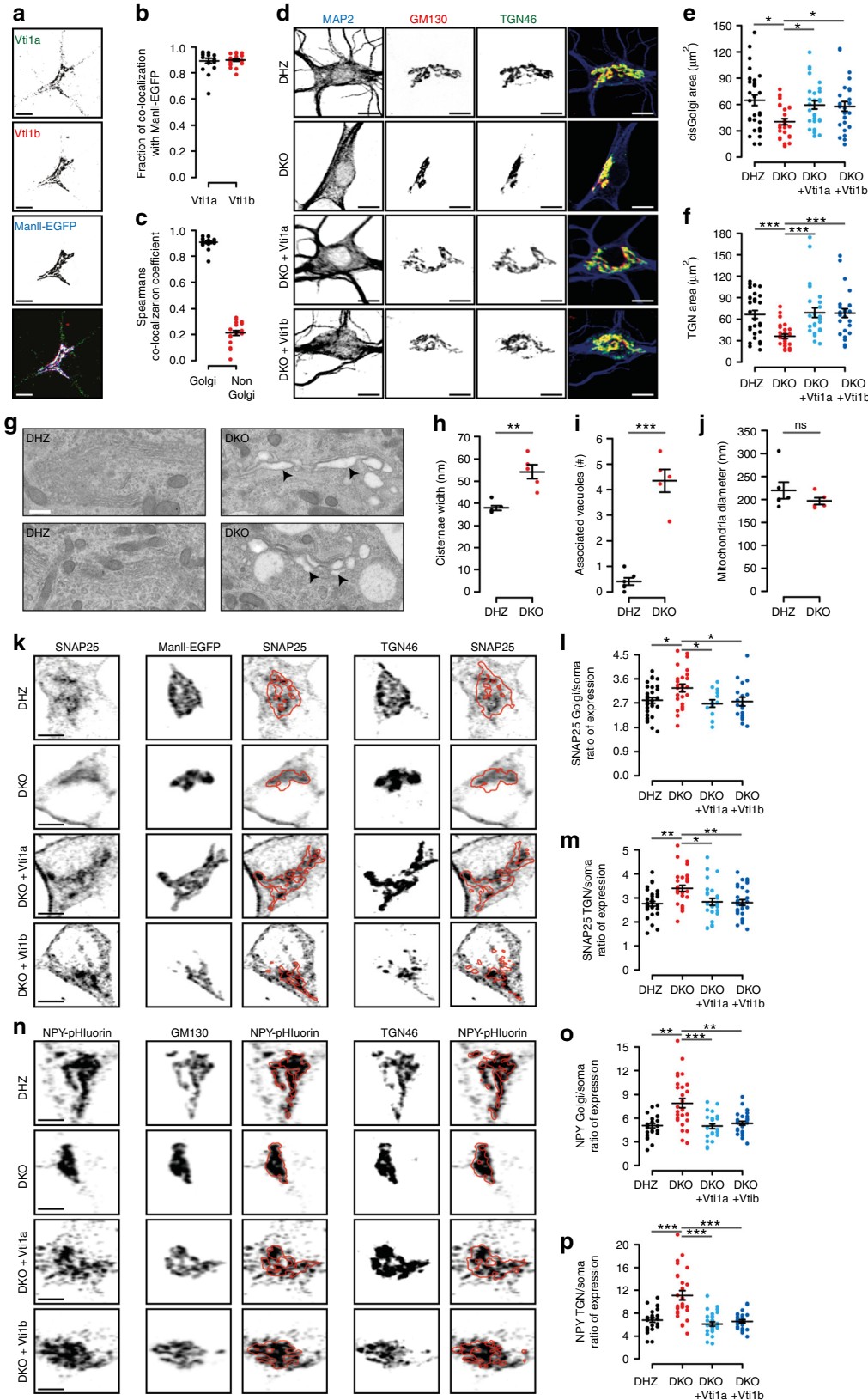

in DKO neurons (Fig. 7d, e) as indicated by the lower fraction that exited the Golgi 64 or 104 minutes after biotin application (Fig. 7g, h). In astrocytes, similar results were obtained (Supplementary Figure 10). For GPI, the export kinetics were initially slower in DKO neurons (32% less cargo exited the Golgi 64 min after biotin application; Fig. 7l), but later converged (Fig. 7m). These data indicate that Vti1a/b are necessary for efficient Golgi export of regulated cargo and, to a lesser extent, constitutive cargo, but not for ER-to-Golgi transport.

**Fig. 6** Golgi morphology abnormalities and protein accumulation in Vti1a/b-deficient neurons. **a** Examples of DIV-14 single DHZ neurons expressing ManII-EGFP and immunostained for Vti1a/b. **b** ManII-EGFP overlaps with Vti1a and Vti1b (89 ± 2 and 90 ± 1%, respectively; $n = 19$). **c** Spearman's correlation coefficient for Vti1a and Vti1b inside and outside of the Golgi (93 ± 1 and 22 ± 2%, respectively; $n = 19$). **d** Examples of DIV-14 single neurons immunostained for MAP2, TGN46 and GM130. **e** Reduced GM130-positive area in DKO neurons (DHZ: 64.60 ± 6.05, $n = 30$; DKO: 40.42 ± 3.49, $n = 29$; DKO + Vti1a: 59.46 ± 4.74, $n = 29$; DKO + Vti1b: 57.91 ± 5.40 $\mu m^2$, $n = 27$; Kruskal–Wallis). **f** Reduced TGN46-positive areas in DKO neurons (DHZ: 67.13 ± 5.46, $n = 30$; DKO: 36.45 ± 2.98, $n = 29$; DKO + Vti1a: 69.38 ± 6.66, $n = 29$; DKO + Vti1b: 68.73 ± 6.01$\mu m^2$, $n = 27$; Kruskal–Wallis). **g** Representative electron micrographs of DIV-14 neurons. Arrowhead and asterisks point at examples of distended Golgi and clear-core vacuoles, respectively. **h** Increased Golgi cisternae width in DKO neurons (DHZ: 37.82 ± 1.00; DKO: 54.24 ± 3.24 nm; Mann–Whitney). **i** More vacuoles within 500 nm of Golgi stacks in DKO neurons (DHZ: 0.41 ± 0.14; DKO: 4.34 ± 0.45 vacuoles; t-test). **j** Similar mitochondria diameter between groups (DHZ: 219.87 ± 18.02; DKO: 196.59 ± 7.63 nm; Mann–Whitney). **k** Examples of DIV-5 single neurons expressing MannII-EGFP and immunostained for SNAP25 and TGN46. Red lines represent MannII-EGFP- or TGN46-positive areas. **l** Increased SNAP25 expression ratio at the Golgi in DKO neurons (DHZ: 2.80 ± 0.10, $n = 31$; DKO: 3.26 ± 0.14, $n = 26$; DKO + Vti1a: 2.68 ± 0.14, $n = 14$; DKO + Vti1b: 2.76 ± 0.16, $n = 18$; ANOVA). **m** Increased SNAP25 expression ratio at the TGN in DKO neurons (DHZ: 2.76 ± 0.10, $n = 33$; DKO: 3.40 ± 0.13, $n = 29$; DKO + Vti1a: 2.84 ± 0.15, $n = 26$; DKO + Vti1b: 2.80 ± 0.12, $n = 26$; ANOVA). **n** Examples of DIV-5 single neurons expressing NPY-pHluorin and immunostained for GM130 and TGN46. Red lines represent GM130- or TGN46-positive areas. **o** Increased NPY-pHluorin ratio of expression at the Golgi in DKO neurons (DHZ: 5.03 ± 0.28, $n = 25$; DKO: 7.88 ± 0.59, $n = 28$; DKO + Vti1a: 4.96 ± 0.30, $n = 27$; DKO + Vti1b: 5.33 ± 0.25, $n = 28$; Kruskal–Wallis). **p** Increased NPY-pHluorin expression ratio at the TGN in DKO neurons (DHZ: 6.82 ± 0.39, $n = 25$; DKO: 11.11 ± 0.79, $n = 28$; DKO + Vti1a: 6.13 ± 0.39, $n = 27$; DKO + Vti1b: 6.56 ± 0.29, $n = 28$; Kruskal–Wallis). Bars show mean ± SEM. Scatterplots and columns represent individual neurons and litters, respectively. *$p < 0.05$; **$p < 0.01$; ***$p < 0.001$. Scale bar = 10 μm (**a, d**), 500 nm (**g**), 5 μm (**j, m**). Expression ration represent reporter intensity in Golgi or TGN over rest of the soma

**Impaired retrograde transport in Vti1a/b DKO neurons**. Because SNARE proteins mediate membrane fusion, the abnormal Golgi structure and cargo export may be the consequence of defective fusion reactions at the Golgi in DKO neurons. Since anterograde transport from the ER was unaltered (Fig. 7), we investigated retrograde transport to the Golgi using Cholera Toxin subunit-B fused to Alexa-488 (CTB-A488) as reporter[30]. In DIV-14 neurons, CTB-A488 expression was predominantly observed at the Golgi 2 h after application (Fig. 8a). This fluorescence was 37% lower in DKO neurons (Fig. 8b) and 27% lower after normalizing to the fluorescence outside this organelle (Fig. 8c, d). Vti1a expression rescued the relative florescence at the Golgi, but Vti1b expression only showed partial rescue (Fig. 8a–d). Because CTB transport relies on synaptic activity[30], we repeated the experiment at DIV-5, but no alterations were observed 2 h after incubation (Supplementary Figure 11). Because we previously observed that defects in GPI export from the Golgi were only detected after short, but not long, incubation times (Fig. 7l–m), we reasoned that a similar time-dependency may affect CTB-A488 and explain the absence of a measurable phenotype 2 h after CTB-A488 incubation in DIV-5 neurons. To explore this possibility, we measured CTB-A488 expression at the Golgi 30 min after incubation and observed that, at this (shorter) time point, the relative fluorescence at the Golgi was decreased in DKO neuron (Supplementary Figure 11A-D).

The mammalian VPS10 orthologs Sortilin and Sorcs1 sort regulated cargo[31–33]. Therefore, we studied these proteins as candidates to explain altered Golgi export in DKO neurons. However, cellular levels and distribution of these proteins were normal (Supplementary Figure 12A-D), and Vti1a/b and Sortilin or Sorcs1 did not co-immunoprecipitate (Supplementary Figure 12, E, F), as opposed to Vti1a/b and a Vti interactor, Syntaxin-16[16], or Vti1p and VPS10 in yeast[34] (Supplementary Figure 12G).

These results indicate that DKO neurons have reduced retrograde transport of CTB-A488, which is fully rescued by Vti1a, and partially by Vt1b. Furthermore, expression or distribution of the VPS10 proteins Sortilin and Sorcs1 provide no explanation for altered Golgi sorting in DKO neurons.

**Discussion**

We investigated the function of Vti1a/b in regulated secretion. We showed that in the absence of Vti1a/b neurons form less synapses,

contain fewer DCVs and have impaired secretion capacity. Furthermore, the synaptic levels of proteins that regulate secretion were lower, the influx of SNAP25 and DCV cargo into the axon was decreased and these molecules accumulated in the Golgi. The expression of either Vti1a or Vti1b rescued all these phenotypes. Finally, Golgi morphology was abnormal, with distended cisternae and clear vacuoles, cargo export from the Golgi was compromised, and retrograde, but not anterograde, trafficking to the Golgi was impaired.

Vti1a and Vti1b are the two mammalian orthologs of the single-yeast *Vti1p* gene[34]. Vti1a and Vti1b share only 31% homology[35]. In vitro and ex vivo studies have reported separate functions for these proteins in distinct steps of intracellular trafficking[15–19]. In line with this, we observed (partial) co-localization of Vti proteins with endo-lysosomal markers. We also found that Vti proteins co-localize at the Golgi and that many Vti1a/b-null phenotypes in neurons were fully rescued by the expression of either Vti1a or Vti1b, indicative of redundant functions. Furthermore, the fact that DKO mice show neurodegeneration and die at birth, while single null mutants are viable and fertile[12,36] supports shared functions for these proteins. Hence, Vti1a and Vti1b support many cellular functions in neurons in an equally efficient and largely redundant manner. Such redundancy may have remained unnoticed because most of the previous studies selectively addressed single Vti proteins and/or used systems where only a single-Vti protein may be expressed. Alternatively, redundancy may be specific for neurons, which is supported by the fact that neurons are the most affected cell type in the DKO mice[12]. The reduced DCV pool and impaired CTB delivery to the Golgi were the only two phenotypes more efficiently rescued by Vti1a than Vti1b. This aligns with the decreased number of secretory granules in Vti1a KO chromaffin cells[24] and the Vti1a-dependent transport of Shiga toxin to the Golgi in cell lines[16]. Hence, although few aspects of intracellular trafficking are more efficiently supported by Vti1a, Vti1a and Vti1b generally support cellular functions in a largely redundant manner, at least in neurons.

Vti1a is detected in SV fractions[14,37] and local roles in endosomal sorting of SVs[38] and spontaneous SV fusion[20,39] have been proposed. This latter proposal was supported by the spontaneous fusion of Vti1a-labeled acidic organelles (which were distinct from Syntaxin6- or transferrin receptor-labeled organelles) and by a reduced spontaneous SV fusion upon single Vti1a or combined Vti1a/VAMP7 knockdown[20,39]. We observed that SV and

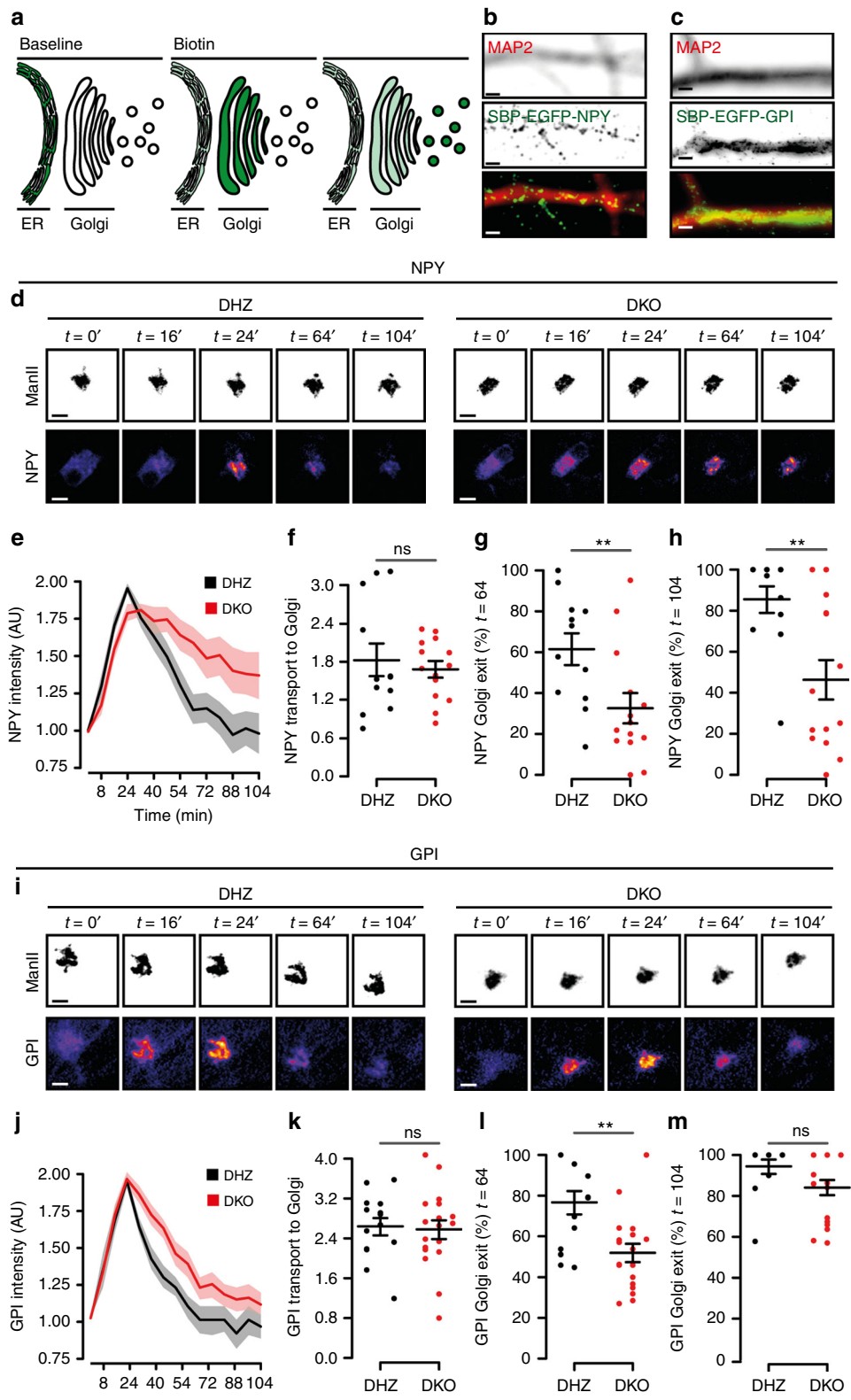

DCV secretion were severely impaired in Vti1a/b-deficient neurons. Because SNAP25 supplies the Qb domain in the SNARE complex that drives SV exocytosis, also spontaneous[22,23], it seems improbable that the Qb-SNAREs Vti1a/b have a direct role in SV fusion with the plasma membrane. Our data demonstrates that Vti1a/b maintain normal levels of, at least, SNAP25, Munc13-1,

RIM, Bassoon, and Synaptotagmin-1 and, to a lesser extent, Syntaxin-1, Synaptobrevin-2, and Munc18-1, and that at least SNAP25 accumulates at the Golgi. Deficiency for most of these proteins inhibits SV fusion[3,40,41] and DCV fusion[21,25,42], and haploinsufficiency also affects synaptic transmission for at least, SNAP25 or Munc18-1[43,44]. Hence, the combined reduction of the

**Fig. 7** Cargo trafficking out of, but not into, the Golgi is reduced in Vti1a/b-deficient neurons. **a** Cartoon representing trafficking assay (originally described by ref. [29]). Fluorescent cargo fused to SBP (Streptavidin-binding protein) is retained in the ER of KDEL-Streptavidin expressing neurons. Biotin reverses SBP-Streptavidin binding and cargo is free to traffick to the Golgi and, subsequently, outside of this organelle. **b**, **c** Representative examples of neurons expressing NPY-SBP-EGFP (**b**) or GPI-SBP-EGFP (**c**) and immunostained with MAP2. **d** Representative time point examples of DHZ (left) and DKO (right) neurons expressing NPY-SBP-EGFP as reporter and ManII-ECFP as Golgi marker. Biotin was added at $t = 0$. **e** Fluorescent traces of NPY-SBP-EGFP in the Golgi. **f** Similar ER-to-Golgi transport of NPY, measured as the fitted slope, between groups (DHZ: 1.83 ± 0.26, $n = 12$; DKO: 1.68 ± 0.13, $n = 14$; $t$-test. **g** Golgi exit of NPY, measured as the percentage of cargo that leaves the Golgi relative to the peak, at $t = 64$ (DHZ: 61.36 ± 7.74, $n = 12$; DKO: 32.61 ± 7.45%, $n = 14$; Mann–Whitney). **h** Golgi exit of NPY, measured as the percentage of cargo that leaves the Golgi relative to the peak, at $t = 104$ (DHZ: 85.49 ± 6.51, $n = 12$; DKO: 46.27 ± 9.59%, $n = 14$; Mann–Whitney). i Representative time point examples of DHZ (left) and DKO (right) neurons expressing GPI-SBP-EGFP and MannII-ECFP as Golgi marker. Biotin was added at $t = 0$. **j** Fluorescent traces of GPI-SBP-EGFP in the Golgi. **k** ER-to-Golgi transport of GPI, measured as the fitted slope, is similar between groups (DHZ: 2.66 ± 0.17, $n = 15$; DKO: 2.57 ± 0.19, $n = 18$; $t$-test). **l** Golgi exit of GPI, measured as the percentage of cargo that leaves the Golgi, 64 min after the ER peak (DHZ: 76.58 ± 5.78, $n = 14$; DKO: 51.87 ± 4.45%, $n = 18$; Mann–Whitney). **m** Golgi exit of GPI, measured as the percentage of cargo that leaves the Golgi, 104 min after the ER peak (DHZ: 94.27 ± 3.66, $n = 12$; DKO: 84.10 ± 3.83%, $n = 18$; Mann–Whitney). Bars and trace plots show mean ± SEM. Scatterplots and columns represent individual neurons and independent litters, respectively. **$p < 0.01$. Scale bar = 2 μm (**b**, **c**) and 5 μm (**d**, **h**)

levels of these proteins provides a plausible explanation for the impaired secretion in the absence of Vti1a/b. Although our data cannot exclude local synaptic functions for Vti1a/b, we conclude that Vti1a/b support synaptic secretion by sorting and targeting components of the secretion machinery to synapses.

Fewer DCVs were observed in synaptic micrographs of DKO neurons and staining for DCV-cargo was reduced, whereas DCV cargo loading per vesicle was unaltered. This indicates that DKO neurons express fewer DCVs with probably unaltered cargo composition. In contrast, the number of SVs per synapse was unaffected, while staining intensity for synaptic markers was decreased in DKO synapses. This suggests that, unlike DCV biogenesis, SV biogenesis is unaffected by the loss of Vti1a/b, despite the reduced levels of synaptic proteins. We conclude that SV biogenesis is coupled to the rate of exocytosis, as suggested[2], and does not depend on Vti1a/b. Furthermore, given the reduced staining of synaptic proteins, SVs most likely contain a reduced number of native proteins per vesicle in DKO neurons. This provides a plausible explanation for the impaired synaptic transmission in Vti1a/b neurons.

The influx of DCV-cargo and proteins of the secretion machinery into the axon was decreased in DKO neurons, but co-localization of these proteins with lysosomes or their cellular stability were unaltered. Therefore, the reduced levels of DCVs and proteins that drive secretion are most likely explained by decreased axonal influx, not by aberrant protein turnover. The two main cellular pathways that can explain the decreased influx into axons are impaired maturation of transport vesicles (by homotypic or heterotypic fusion) or impaired protein export from the Golgi. The Vti partner Syntaxin-6 mediates homotypic fusion of secretory organelles ex vivo[9], arguing for the first scenario. However, the somatic distribution of DCVs and the cargo content of DCVs were normal in Vti1a/b-deficient neurons, suggesting unaffected maturation. A different Qb-SNARE may support vesicle maturation or, alternatively, this process may not be essential in neurons. In contrast, we observed independent lines of evidence for the second scenario, impaired Golgi export. The Golgi was smaller and cisternae were abnormal, the normal non-homogeneous distribution of DCV-cargo inside the Golgi was lost, cargo accumulated at the Golgi and the transit time of labeled cargo through this organelle was abnormally long. In addition, the export of labeled cargo was also delayed in astrocytes, indicating that Vti1a/b do not only support this function in neurons. Therefore, we conclude that Vti1a/b are required for normal protein export from the Golgi.

The accumulation of DCV-cargo in the Golgi and its abnormally long transit time are consistent with the reduced number of secretory granules in Vti1a-null chromaffin cells[24] and the

accumulation of tumor necrosis factor-α in a Golgi-like compartment in macrophages after Vti1b inhibition[45]. Our study shows that targeting of cargo presumably not sorted to the regulated secretory pathway, like SNAP25, GPI, PSD95, and TrKB receptors, was also impaired in DKO neurons. This is in line with the impaired delivery to the plasma membrane of $Ca^{2+}$ channels in Vti1a-null chromaffin cells[24] and Kv4 $K^+$ channels in HeLa cells deficient for Vti1a or the Vti SNARE partner VAMP7[46], and indicates that the exit of many proteins from the Golgi relies on Vti1a/b.

Neurons degenerate in the absence of Vti1a/b[12]. Because neuronal death was already abundant at DIV-4, when regulated secretion is minimal[21], and because abolishing regulated secretion perse does not trigger neuronal death[47], the degeneration observed in DKO neurons is probably not a direct consequence of impaired exocytosis. However, we cannot exclude that the combined loss of (many) membrane proteins contributes to the degeneration observed. Despite the accelerated cell death of DKO neurons, many cellular functions were normal. The distribution of synapses and DCVs was unaffected, DCV trafficking speed (around 1 μm/s) was in the range of reported values[26,48]; and SV endocytosis, calcium influx and efflux and vesicular acidification were similar to DHZ neurons, confirming that Vti1a/b DKO neurons can cope with energy-demanding tasks. Furthermore, protein stability, mitochondrial morphology and ER-to-Golgi cargo trafficking were unaffected. This is in line with previous observations that many intracellular trafficking pathways, including recycling of transferrin receptors, delivery of endocytosed cargo to lysosomes and autophagy, were fully functional in Vti1a/b DKO fibroblasts[12]. Golgi abnormalities in the absence of Vti1a/b were also observed in acute brain slices, indicating that this is not a feature unique for cultured neurons. Furthermore, the levels of synaptic proteins and SV fusion were rescued by expression of Vti1a or Vti1b after most DKO neurons died, which confirms that these features are not intrinsic of the surviving neurons. Hence, we conclude that loss of Vti1a/b function does not trigger generalized cellular dysfunctions, but selectively affects regulated secretion by impaired Golgi export of relevant proteins, which is not a unique feature for surviving neurons or culture conditions.

Molecules are delivered to the Golgi anterogradely from the ER, or retrogradely from the plasma membrane or intracellular organelles[49]. ER-Golgi trafficking was normal in Vt1a/b-deficient neurons and astrocytes, in line with the fact that other Qb-SNAREs mediate this anterograde pathway[50,51]. In contrast, retrograde transport from the plasma membrane to the Golgi was impaired in Vti1a/b-deficient neurons, consistent with the decrease in Shiga toxin transport in cell lines upon interference

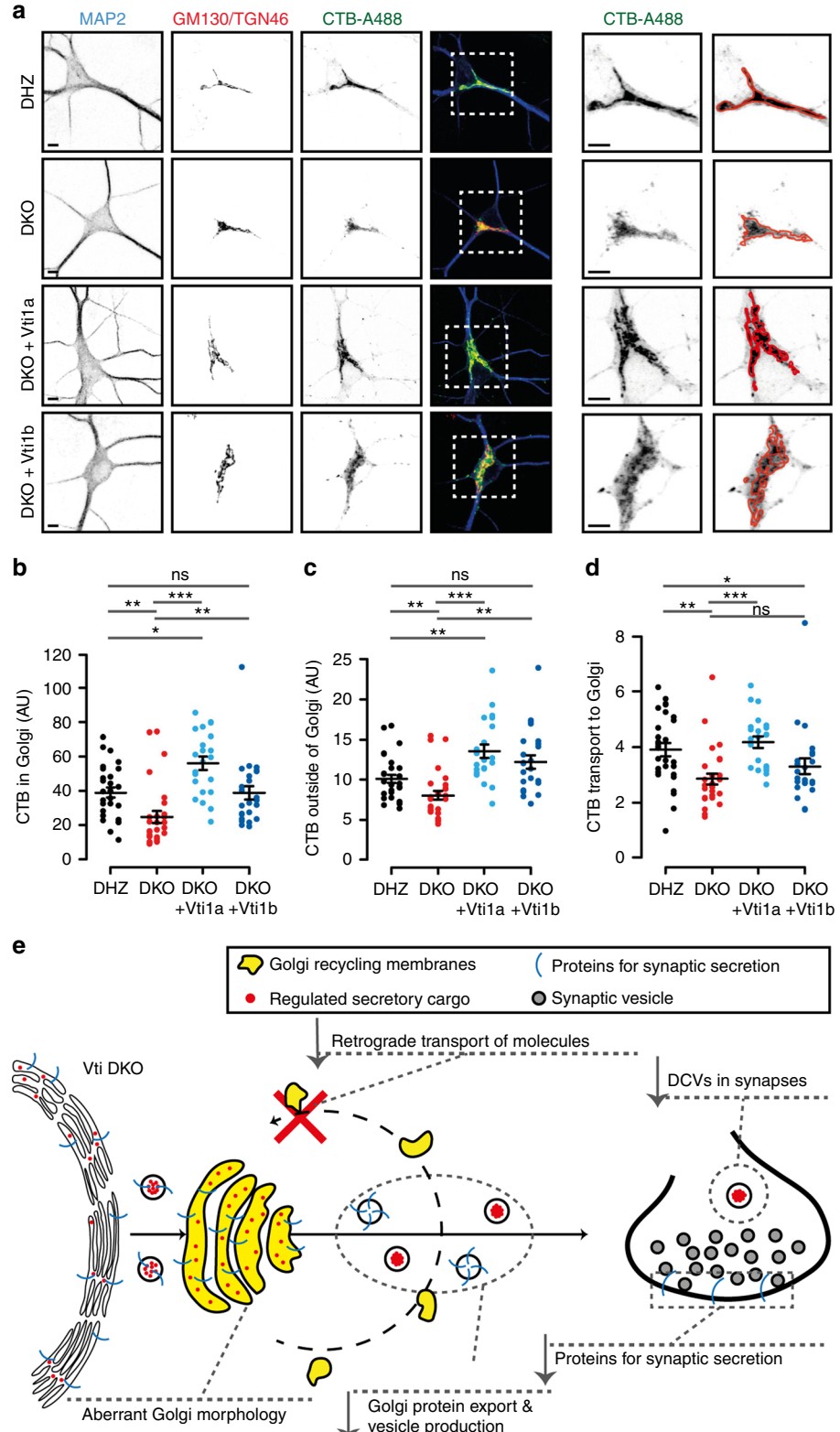

with Vti1a-function using antibodies[16]. The preferential localization of Vti1a/b to the Golgi aligns with a role in retrograde transport to this organelle. Vti1b localized to the Golgi in neurons to a larger extent than what was reported in fibroblast cell lines[52], which may indicate cell type differences. Vti1a/b deficiency had a larger impact on retrograde transport at DIV-14 than at DIV-5. This can generally be explained by a progressive nature of this

defect: components that regulate this trafficking probably become progressively missorted over time. Alternatively, different Qb-SNARE may be involved in similar pathways during different developmental stages[21], or synaptic activity influences CTB retrograde trafficking, as suggested[30]. Expression of either Vti1a or Vti1b in DKO neurons increased CTB delivery to the Golgi, but only Vti1a fully rescued it. Retrograde transport of Shiga toxin to

**Fig. 8** Defective retrograde transport in Vti1a/b-deficient neurons. **a** Representative examples and zooms of DIV-14 single neurons incubated with CTB-A488 and immunostained for MAP2, GM130, and TGN46. Red lines represent GM130/TGN46-positive areas. Scale bar = 5 μm. **b** CTB-A488 intensity at GM130/TGN46-positive areas (DHZ: 39.01 ± 2.88 $n = 29$; DKO: 24.73 ± 3.39, $n = 29$; DKO + Vti1a: 56.11 ± 3.90, $n = 22$; DKO + Vti1b: 38.901 ± 4.03 AU, $n = 24$; Kruskal–Wallis). **c** CTB-A488 intensity, excluding GM130/TGN46-positive areas (DHZ: 100.80 ± 5.01, $n = 29$; DKO: 80.05 ± 5.59, $n = 29$; DKO + Vti1a: 135.47 ± 8.32, $n = 22$; DKO + Vti1b: 122.03 ± 8.19 AU, $n = 24$; Kruskal–Wallis). **d** CTB-A488 transport to the Golgi, measured as intensity in the Golgi over rest of the neuron, is reduced in DKO neurons and only rescued by Vti1a (DHZ: 3.90 ± 0.23, $n = 29$; DKO: 2.85 ± 0.19, $n = 29$; DKO + Vti1a: 4.17 ± 0.21, $n = 22$; DKO + Vti1b: 3.28 ± 0.28, $n = 24$; Kruskal–Wallis). **e** Original working model. Impaired retrograde transport of membrane and sorting molecules to the Golgi in DKO neurons results in a decreased Golgi size, aberrant Golgi ultrastructure and reduced cargo export which, ultimately, compromises regulated secretion. Bars show mean ± SEM. Scatterplots and columns represent individual neurons and litters, respectively. *$p < 0.05$; **$p < 0.01$; ***$p < 0.001$

the Golgi depends on EpsinR[53], a Vti1b interactor[54], which confirms the relevance of Vti1b in plasma membrane-to-Golgi transport. In addition to plasma membrane–Golgi transport, additional retrograde transport routes may be affected in DKO neurons. Indeed, Vti1a mediates late endosome-to-Golgi transport[17], also in yeast[55]. Defective transport of sorting molecules between endo-lysosomal organelles and the Golgi may also help to explain Golgi export defects in Vti1a/b DKO neurons, and would fit with the role of endo-lysosomal proteins in sorting of regulated secretory cargo[10,56,57].

Several cellular phenotypes show striking similarities to those observed for Vti1a/b deficiency. First, blocking expression of the Golgi-Associated Retrograde Protein (GARP)-complex subunit Vps54 in cell lines resulted in inefficient transit of GPI from the Golgi to the cell surface[58]. Second, disrupting either the Conserved Oligomeric Golgi (COG) or the GARP tethering complexes, which function in recycling to or within the Golgi, resulted in distended Golgi cisternae and vacuolization[53,59–61]. These similarities suggest that Vti1a/b may work together with one or both these complexes to regulate transport to or within the Golgi and that impaired delivery of molecules that control protein sorting may explain the cellular phenotypes described in our study (Fig. 8e). Interestingly, Vti1a indeed interacts with COG and GARP subunits[62,63], but the functional implications of these interactions remains unknown.

In yeast, Vti1p binds to the sorting receptor Vps10 and a Vti1p mutation causes cargo missorting[34]. In mammals, the VPS10 receptors Sortilin and Sorcs1 bind and sort regulated cargo[31–33,37–39], and would provide a plausible link between retrograde trafficking defects and other cellular phenotypes observed in Vti1a/b DKO neurons. However, the levels and distribution of these receptors were unaltered in DKO neurons, and binding to mammalian Vti1a or Vti1b was not detected. Additional molecules have been implicated in protein sorting at the Golgi, including calcium transporters[64], proton pumps[65], HID-1[66], CAB45[67], and lipids[68,69]. Cytosolic proteins, such as actin regulators[64], adapter complexes[56], enzymes[70], and Arf members[71,72], also contribute to Golgi sorting, but the correct localization of these soluble proteins probably does not rely directly on Vti1a/b-dependent membrane trafficking. An overarching model for protein sorting at the Golgi is lacking and the link between molecules that function in this process and Vti1a/b remains unclear. However, the current study clearly indicates that Vti1a/b function is central to coordinate Golgi import/export, with major downstream consequences if this aspect is not functional.

## Methods

**Laboratory animals and primary cultures**. Vti1a/1b null mice, generated by replacing a 6.5 Kb fragment in Vti1a exon 6 and a 7 Kb fragment in Vti1b exon 4 with neomycin resistance cassettes, have been described before[12,36]. Embryonic day (E) 18.5 embryos were obtained by cesarean section of pregnant females from timed mating of Vti1a± and Vti1b ∓ −/+ mice, or vice versa. Mouse hippocampi were dissected from these embryos in HBSS (Sigma), digested in 0.25% trypsin (Life technologies) for 15 min at 37 °C and dissociated with fire-polished Pasteur

pipettes. Dissociated neurons were resuspended in Neurobasal supplemented with 2% B-27, 18 mM HEPES, 0.25% Glutamax, 0.1% penicillin/streptomycin (Life technologies), and plated on pre-grown glia cells (25,000 to 50,000 neurons/well for DHZ and DKO, respectively). For astrocyte micro-islands, 1500 or 4000 neurons/well for DHZ and DKO, respectively, were plated. A higher density was used for DKO to compensate for the neuronal death. DHZ were used as controls due to the low probability to obtain DKO and double-WT mice within the same litter. Neurons were maintained at 37 °C, 5% $CO_2$ for 13–15 days (referred as DIV-14) or 4–6 days (referred as DIV-5).Rat and mouse astrocytes were cultured from P1-2 pup cortices. For functional experiments, astrocytes were plated in T75 flasks, grown in DMEM with 10% FCS, 1% NEAA, 1% penicillin/streptomycin (Life Technologies) at 37 °C, 5% $CO_2$ till confluent, and subsequently replated on glass coverslips and grown for 1 week. For micro-islands, astrocytes were prepared by plating 6000 rat glia cells per agarose-coated glass coverslip stamped with 0.1 mg/ml poly-D-lysine (Sigma) and 0.7 mg/ml rat tail collagen (BD Biosciences). Animals were housed and bred according to institutional and Dutch guidelines.

**Plasmids**. Vti1a, Vti1b, NPY-pHluorin, NPY-mCherry, Synapsin-ECFP, and Synaptophysin-pHluorin plasmids have been described[24–27,36]. NaV$_{II/III}$-mCherry was engineered by substituting YFP by mCherry in NaV$_{II/III}$-YFP (Addgene, 26056). ManII-ECFP was made by substituting GFP by ECFP in ManII-GFP (gift from V. Malhotra, Centre for Genomic Regulation, Barcelona, Spain). The above plasmids were cloned into pLenti vectors under the human Synapsin promoter and viral particles were delivered to neurons at DIV-1 or DIV-10. For RUSH experiments, CMV driven Streptavidin-KDEL/SBP-EGFP-NPY was engineered substituting GPI from Streptavidin-KDEL/SBP-EGFP-GPI (Addgene, 65294). Standard calcium phosphate precipitation was used to deliver these plasmids, in combination with ManII-EGFP. Cells were analyzed 20–28 h after transfection.

For immunoprecipitation, Vti1a and 1b were fused to Myc tags, and SorCS1 (gift from J. de Wit, VIB Center for the Biology of Disease, Leuven, Belgium) and Sortilin (gift from C. M. Petersen, Aarhus Universiteit, Denmark) were fused to Flag tags. These plasmids were expressed under CMV promoters and transfected into HEK293T cells.

**Immunostaining**. Neurons, fixed in 3.7% paraformaldehyde (EMS) for 20 min, were permeabilized with 0.5% Triton-X (Fisher Scientific) for 5 min and blocked with 2% normal goat serum (Life Technologies) and 0.1% Triton-X for 20 min. Incubation with primary and secondary antibodies was done at room temperature (RT) for 2 or 1 h, respectively. All solutions were in PBS (composition in mM: 137 NaCl, 2.7 KCl, 10 Na$_2$HPO$_4$, 1.8 KH$_2$PO$_4$; pH = 7.4). Coverslips were mounted in Mowiol (Sigma) and Z-stacks were obtained on a confocal microscope (Nikon Eclipse Ti) equipped with ×10 (NA = 0.45) and ×63 oil (NA = 1.40) objectives controlled by NisElements 4.30 software. For retrograde labeling experiments, neurons were incubated at 37 °C with CTB-A488 (Thermo Fisher; 100 ng/ml) for 15 min and fixed 30 min or 2 h after incubation. For protein stability, neurons were incubated with cycloheximide (Sigma, 100 μg/ml) in 0.1% DMSO, or 0.1% DMSO alone, for 24 h before fixation.

Primary antibodies used for immunocytochemistry: Vti1a (1:1000, BD, 611220), Vti1b (1:500[14]), MAP2 (1:1000, Abcam, ab5392), Munc18-1 (1:1000, BD, 610336), SNAP25 (1:1000, Abcam, SMI81), Synaptobrevin-2 (1:1000, SySy, mouse), Syntaxin-1 (1:1000, I379, gift from T. C. Südhof, Stanford University), Synaptophysin-1 (1:1000, SySy, 101004), Munc13-1 (1:1000, SySy, 126 111), Synaptotagmin-1 (1:1000, W855, gift from T. C. Südhof, Stanford University), RIM (1:1000, SySy, 140203), Bassoon (1:1000, Enzo, SAP7F407), GM130 (1:1000, BD, 610822), TGN46 (1:1000, Abcam, ab76282), SMI312 (1:1000, Covance, SMI312R), VGLUT-1 (1:5000, Millipore, ab5905), VGAT (1:1000, SySy, 131002), Lamp1 (1:1000, Abcam, ab25245), BDNF (1:8, developed by Yves-Alain Barde, (Biozentrum University of Basel, and obtained from the Developmental Studies Hybridoma Bank), TrKB (1:500, Millipore, 07-225), PSD95 (1:250, SySy. 124 011), Rab5 rabbit (1:1000, Abcam, ab18211), and Rab5 mouse (1:100, SySy, 108 101). Secondary AlexaFluor-conjugated antibodies (1:1000, Invitrogen).

For acute slices, E18.5 brains were fixed in Bouin's solution and embedded in paraffin. Sagittal thick sections (7 μM) were obtained and deparaffinized. Deparaffinized sections were renatured by microwaving for 5 min in boiling 0.1 M

citrate buffer (pH 6) and incubated with 1% normal goat serum in PBS for 1 h. Renatured sections were incubated with primary mouse anti-GM130 antibody (1:400 in PBS, 1% goat serum) overnight at 4 °C, followed by Cy3-cojugated anti-mouse antibodies (Jackson ImmunoResearch, 1:400 in PBS with 1% goat serum) for 1 h at RT. Nuclei were stained in Hoechst dye (1:1000; Thermo Fisher, 33342).

**Live imaging.** Coverslips were placed in imaging chambers perfused with Tyrode's buffer (composition in mM: 2 CaCl$_2$, 2.5 KCl, 119 NaCl, 2 MgCl$_2$, 20 Glucose, 25 HEPES, pH 7.4) and recorded at RT using a Zeiss AxioObserver.Z1 (SV and DCV fusion) or at 37 °C (FRAP and RUSH) using a Nikon Eclipse Ti microscope. Microscopes were equipped with laser-based illumination (444, 488, and 561 nm), appropriate filter sets, ×40 oil (NA = 1.3) or ×63 oil (NA = 1.4) objectives and an EM charge-coupled device camera (C9100-02, Hamamatsu). Time-lapse recordings were acquired using Axiovision 4.8 or NisElements 4.30. The acquisition frequency was 2 Hz for DCV fusion, FRAP and calcium imaging; 0.6 Hz for dual color (SypHy/FM4-64) SV fusion and 1 Hz for single (SypHy) SV fusion rescued at DIV10. HFS was applied using parallel platinum electrodes delivering 30 mA, 1 ms pulses controlled by a Master-8 (AMPI) and a stimulus generator (A-365, WPI). Intravesicular pH was neutralized by gravity flow application of modified Tyrode's solution (50 mM NaCl replaced by 50 mM NH$_4$Cl). For FRAP experiments, laser intensity and pulse duration for bleaching were optimized to reach >90% fluorescence decrease of NPY-mCherry or SNAP25-EGFP at the axon initial segment (labeled using NaV$_{II/III}$-mCherry or -EYFP). For calcium imaging, neurons were incubated with 2 µM Fluo5F AM (Molecular Probes) in supplemented Neurobasal at 37 °C, 5% CO$_2$ for 10 min. For RUSH experiments[29], neurons were imaged in supplemented Neurobasal. Cycloheximide (Sigma, 100 µg/ml) and Biotin (Sigma, 40 µM) were added at recording time-0. FM4-64 (Thermo Fisher; 2 µM) was dissolved in Tyrode's containing 60 mM K$^+$ and incubated for one minute, followed by a 10-min wash, before imaging.

**Image analysis.** A custom-written algorithm in MATLAB (Mathworks), SynD[73], was used to quantify dendritic and axonal length, and SV and DCV puncta from maximum projections of confocal images. For SV and DCV fusion analysis, traces were expressed as fluorescence change ($\Delta F$) over initial fluorescence ($F_0$, obtained by averaging the first 10-s frames). For SV fusion, 4×4 pixel ROIs were placed on all synapses, identified as non-mobile SypHy puncta with a $\Delta F/F_0 > 3$ standard deviation (sd) upon NH$_4^+$ superfusion. For DCV fusion, 3×3 pixel ROIs were manually placed on all NPY-pHluorin puncta that appeared upon HFS stimulation. Only when $\Delta F/F_0 > 3$ sd, ROIs were counted as DCV fusion events. The total DCV pool was quantified from the maximum projection during NH$_4^+$ superfusion. SynD was used for DCV puncta detection. The detected puncta may contain several DCVs due to resolution limitations and, hence, pool underestimation was avoided by applying puncta intensity correction. Intensity correction considers that the modal puncta intensity per neuron corresponds to the intensity of individual DCVs that fuse, as demonstrated[25]. For synaptic DCV events, binary masks were built in ImageJ (National Institutes of Health) based on Synapsin-ECFP. DCV fusion events with >50% pixels within the binary mask were considered synaptic. For FRAP experiments, the number and speed of NPY-mCherry or SNAP25-EGFP puncta that entered the axon from the soma were calculated using kymographs (ImageJ). For visualization purposes, brightness, and contrast of representative examples was adjusted in a linear and equal fashion between groups.

**Immunoprecipitation and Western blot.** For expression of Vti1a and Vti1b, DIV-14 neurons (cultured in absence of glia) were washed in PBS, homogenized in Laemmli buffer (2% SDS, 10% glycerol, 0.26 M β-mercaptoethanol, 60 mM Tris-HCl, and 0.01% Bromophenol Blue; pH 6.8) and analyzed by western blot. For immunoprecipitations (IPs), HEK293T cells were transfected with Flag- and Myc-tagged proteins and lysed in IP buffer (50 mM Tris-HCl (pH 7.5), 1% TX-100, 1.5 mM MgCl$_2$, 5.0 mM EDTA (pH 8), 100 mM NaCl) 48 h after transfection. Samples were incubated with antibodies against Flag (0.3 µl/IP, Sigma, F1804) or Myc tags (0.5 µl/IP, GeneTex, GTX75953), and protein-A agarose beads (Vector Laboraties). Beads were collected upon centrifugation (16,000×g) and washes with buffers (50 mM Tris-HCl (pH 7.5), 0.1% TX-100, 1.5 mM MgCl$_2$, 5.0 mM EDTA (pH 8)) containing 100 or 200 mM NaCl. Finally, beads were dried using an insulin needle, dissolved in Laemmli Sample buffer, boiled, and analyzed by western blotting.

For Western blots, samples were separated on 10% SDS-polyacrylamide gels and transferred to Nitrocellulose or PVDF membranes. Blots were blocked in 2% milk and 0.5% bovine serum albumin (BSA) for 1 h at 4 °C and incubated with primary and secondary antibodies at 4 °C overnight and 1 h at RT, respectively. Blots were scanned using a Fuji Film FLA 5000. Blocking and antibody solutions were prepared in PBS containing 0. 1% Tween-20.

**Electron microscopy.** Neurons were fixed for 90 min at RT with 2.5% glutaraldehyde in 0.1 M cacodylate buffer, pH 7.4. Cells were washed and postfixed for 1 h at RT with 1% OsO$_4$/1% KRu(CN)$_6$. After dehydration through a series of increasing ethanol concentrations, cells were embedded in Epon and polymerized for 48 h at 60 °C. Ultrathin sections (80 nm) were cut parallel to the cell monolayer, collected on single-slot formvar-coated copper grids, and stained in uranyl acetate and lead citrate in Ultra stainer LEICA EM AC20. Neuronal sections were randomly selected at low magnification and imaged using a JEOL1010 transmission electron microscope at 60 kV.

**Electrophysiology.** Whole-cell voltage-clamp recordings (Vm = −70 mV) were performed at RT with borosilicate glass pipettes (2–5 MΩ) filled with (in mM) 125 K$^+$-gluconic acid, 10 NaCl, 4.6 MgCl$_2$, 4 K$_2$-ATP, 15 creatine phosphate, 1 EGTA, and 10 units/mL phosphocreatine kinase (pH 7.30). External solution contained in mM: 10 HEPES, 10 glucose, 140 NaCl, 2.4 KCl, 4 MgCl$_2$, and 2 CaCl$_2$ (pH = 7.30, 300 mOsmol). Inhibitory neurons were identified and excluded based on the decay of postsynaptic currents. Recordings were acquired with a MultiClamp 700B amplifier, Digidata 1440 A, and pCLAMP 10.3 software (Molecular Devices). Only cells with an access resistance < 15 MΩ (80% compensated) and leak current of <300 pA were included. EPSCs were elicited by a 0.5 ms depolarization to 30 mV. RRP size was assessed by hypertonic sucrose application[74] (500 mM for 7 s), using a piezo-controlled barrel application system (Perfusion Fast-Step, Warner Instruments) and fitted as reported[75]. Offline analysis was performed using custom-written software routines in Matlab R2016b (Mathworks)In all figures, stimulation artifacts have been blanked out.

**Statistics.** Shapiro and Levene's were used to test distribution normality and homogeneity of variances, respectively. When assumptions of normality or homogeneity of variances were met, parametric tests were used: t-test or ANOVA (Tukey as post hoc). Otherwise, non-parametric tests were used: Mann–Whitney U or Kruskal–Wallis test (Holm as post hoc). R was used as software. No pre-determined sample size or randomization were used. No data was excluded. Data was analyzed blindly.

**Code availability.** The custom code used for data analysis during the current study is available from the corresponding author on reasonable request.

**Data availability.** The data sets generated and analyzed during the current study are available from the corresponding author on reasonable request.

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

## Acknowledgements

We thank Robbert Zalm for cloning and producing viral particles, Frank den Oudsten and Desiree Schut for producing glia feeders and primary culture assistance, Joke Wortel for animal breeding, Frank den Oudsten and Joost Hoetjes for genotyping, Rien Dekker and Ingrid Saarloos for expert assistance in electron microscopy and immunoprecipitation studies, respectively, and members of the CNCR DCV project team and Jakob Sorensen (U Copenhagen), Alexander Walter (FMP, Berlin), Rainer Pepperkok (EMBL, Heidelberg) and Judith Klumperman (U Utrecht) for fruitful discussions. This work is supported by an ERC Advanced Grant (322966) of the European Union (to M.V.).

## Author contributions

J.E.M., R.F.T., and M.V. designed the experiments. J.E.M. collected and analyzed experimental data, except for electrophysiology (V.H) electron microscopy (J.v.W.) and brain slice stainings (C.B.). G.F.v.M. facilitated the mouse line. J.E.M, R.F.T., and M.V designed figures and wrote the manuscript with input from all authors.

## Additional information

**Competing interests:** The authors declare no competing interests.

