## [Peer Review File · Nature Communications]

Reviewers' comments:

Reviewer #1 (Remarks to the Author):

The authors provide here a very complex and very thorough analysis of neurons lacking the SNAREs Vti1a and Vti1b. These molecules have been to some extent analyzed in the neuronal context. For example, Vti1a has been proposed to be involved in the exo- and endocytosis of a spontaneously fusing pool of synaptic vesicles (Ramirez et al., 2012; the authors' reference 26). This is a puzzling observation, as the authors also note in the current manuscript, and it does not fit with the main lines of the synaptic transmission field. It is much more likely that Vti1a and Vti1b, which are presumably found on cellular endosomes, are involved in the biogenesis of synaptic elements, and that their effects on synaptic transmission are not related to a direct involvement in the exo- or endocytosis of synaptic vesicles.

The current study provides a much needed perspective in the analysis of SNARE molecules in neurons: the authors study not only the end-point effects on synaptic transmission, but also the influence of these molecules on multiple other processes, including different steps of the secretory pathway. They conclude that Vti1a and Vti1b have largely overlapping functions in the secretory pathway, and that they participate in the biogenesis of synaptic vesicles and of dense-core vesicles, at a step of material recycling in the Golgi apparatus.

I find the current study convincing, well designed, and well executed. As the topic is interested, as noted above, I suggest that the manuscript be published. However, one important point, and a few minor ones, should be explored by the authors before publication.

Major point: the authors should perform a few simple experiments in non-neuronal cells lacking Vti1a and Vti1b, in order to test whether the effects described here are specific for neurons. Could the authors test, for example, retrograde Cholera Toxin trafficking in cultured fibroblasts derived from their mice? Ideally, the authors could also repeat the assay presented in Figure 7, in fibroblasts. Such experiments will test whether the authors' main conclusion is valid in all cells, or only in neurons. Either conclusion would be important, and would strengthen the current manuscript.

Minor points:

- The authors should perform some conventional immunostaining experiments for native Vti1a and Vti1b, in normal neurons, to indicate their rough cellular localizations. Even better, the colocalization with Golgi and endosomal markers could be analyzed.
- The authors may want to analyze the significance of the changes shown in Figure 4. As it stands, some of the changes are very likely significant, as for SNAP25, but this is less clear for other proteins, such as Syntaxin 1 or Munc18-1.
- The authors should include in their discussion an additional paper on presumed synaptic effects of Vti1a: "Selective molecular impairment of spontaneous neurotransmission modulates synaptic efficacy. Crawford DC, Ramirez DM, Trauterman B, Monteggia LM, Kavalali ET. Nat Commun. 2017 Feb 10;8:14436. doi: 10.1038/ncomms14436".

Reviewer #2 (Remarks to the Author):

In this manuscript, Emperador-Melero et al. examined the impact of Vti1a and Vti1b double knockout (DKO) on neuronal survival and morphometric development, trafficking/sorting of neuronal release machinery, and exocytosis of synaptic and dense core vesicles (SVs and DCVs, respectively) using interdisciplinary methodologies including mouse genetics, biochemistry, electron microscopy, electrophysiology and live cell imaging. The authors provide evidence that Vti1a and Vti1b are essential for maintaining neuronal levels of SVs and DCVs and their associated release machineries. Moreover, they show that SNAP25 and DCV cargo likely accumulate in the secretory pathway at the level of the Golgi apparatus likely due to defective retrograde transport.

The paper is generally well-written, and the data will be of interest to the neuroscience community as the essential mediators of protein sorting for maintaining regulated exocytosis in neurons are incompletely understood. Collectively, these data show that Vti1a and Vti1b likely play an important role in maintaining synaptic and other vesicle pools with profound consequences on neuronal development, survival and synaptic function. Nevertheless, while the analyses provided are fairly comprehensive, I would suggest several improvements to strengthen and clarify the conclusions and overall interpretation of the manuscript, as detailed below:

Major points:

1. Although relatively comprehensive characterizations were performed and verified by independent and complimentary methodologies, the main message from this paper is somewhat unclear given the generality and widespread nature of the phenotypes (i.e., cell death, reduced axonal and dendritic maturation, impaired vesicle biogenesis, as well as neurotransmitter and dense core vesicle release). We can generally conclude that Vti1a/b are important for neuronal/synaptic function, survival, and maintenance of secretory vesicle pools, but mechanistic conclusions about Vti1a/b and the causal molecular impairments associated with these phenotypes are somewhat lacking. Some clarification and discussion will be very helpful.

2. It is unclear whether other regulated secretion is impacted by loss of Vti1a/b. For example, are other synaptic membrane proteins (e.g., cell adhesion proteins, ion channels, g-protein coupled receptors, neurotransmitter receptors, etc.) also reduced in Vti1a/b DKO neurons? Are the gross neurodegenerative phenotypes a direct consequence of impaired exocytosis and/or loss or misregulation of other unmeasured proteins that also undergo regulated secretion to the plasma membrane? This will also clarify the specificity of the sorting function by Vti1a/b in neurons.

3. Why is synaptic vesicle number/synapse (EM data, Figure 1K) not impaired since synaptic vesicle biogenesis is also dependent on protein sorting from the Golgi and fluorescence intensity of synaptic markers is markedly decreased in Vti a/b DKO. Moreover, DCV number/synapse is reduced by both EM and immunostaining experiments. Do the authors conclude that these vesicles possess fewer copies of protein per vesicle (as suggested by their western-blotting and immunostaining results)? If so, how do they speculate that normal vesicle numbers are maintained? The authors should provide a clear interpretation/discussion on why levels of synaptic vesicle, but not DCV, number is impacted within the overall interpretation of the study.

Minor points:

1. The statement that "Unlike SVs, DCVs do not locally recycle and, hence, the availability of DCVs depends on the delivery of new vesicles, generated at the trans-Golgi" is a bit far reaching. Whether DCV exocytosis exhibits plasticity or and whether DCVs are locally recycled is not well established.

2. It is unclear why the authors emphasize SNAP25 trafficking and targeting when levels of many other proteins associated with the secretory machinery also appear to be significantly reduced. Simply the fact that SNAP25 levels appear more greatly reduced and they can observe slowed transport of SNAP25 into axons, does not establish that SNAP25 is the major limiting factor in the impairment of exocytosis or other phenotypes. The emphasis should be clarified in the discussion and overall interpretation.

3. In the discussion, the comments that "Vti1a is detected in SV fractions and local synaptic functions have been proposed"..."Because SNAP25 supplies the Qb domain in the canonical SNARE complex that drives SV fusion...it is unclear which domain the Qb SNARE Vti1a contributes" and "these indications that Vti proteins support synaptic secretion are generally confirmed in the current study" are, together, a bit confusing. The authors should probably rephrase this section, as it is unlikely they can draw any specific conclusions regarding a direct role for Vti1 proteins in vesicle exocytosis or recycling based on the data shown here.

4. Figure 1B, is synapse formation redundantly regulated by both Vti1a and Vti1b? The number and intensity of synaptophysin puncta appear only partially rescued by re-expression of either Vti1a or Vti1b, whereas DCV (NPY-pH) number appears fully-rescued by both. Can the authors

double-check their statistics on these data?

5. The mean and range for DCV diameters (60nm and 45-120nm, respectively) (Figure 1O&P) are somewhat small for DCVs (typical range 90-250nm, according to literatures). Can the authors double check these quantifications? What are the average sizes of SVs? Note here I am not asking for quantification for SVs, just want to make sure that the SV and DCVs sizes are different and in general agreement of the literature.

6. What consequences will Vti1a/b DKO have on SV and DCV recycling at their release sites? Can the authors show how SV and DCV releasing sites function in the steady-state during their stimulus trains? If the authors claim that "Unlike SVs, DCVs do not locally recycle and, hence, the availability of DCVs depends on the delivery on new vesicles, generated at the trans-Golgi", they should demonstrate differences in the temporal dynamics of release and maintenance of releasing pools during steady-state stimulations in Vti1a/b DKO

7. Related with major point #2, can the authors provide more analyses to show normal ultrastructural development of synapses in Vti1a/b DKO including the number of docked vesicles and PSD width?

8. Please fix typo in Figure 8 E Vti DKO schematic - "Proteins for synaptic secretion"

Reviewer #3 (Remarks to the Author):

This manuscript reports that Qb-SNAREs Vti1a and Vti1b impact the number of synapses and the levels of synaptic proteins by regulating the sorting and export of proteins from the Golgi. In cultured hippocampal neurons from Vti1a/b double knockout neurons, they find fewer synapses and lower levels of synaptic release machinery proteins. They see fewer dense core vesicles but not synaptic vesicles. Electrophysiological and imaging experiments suggest both synaptic vesicles and dense core vesicles are less likely to be released in response to stimulation. The export of cargo from the Golgi appears to be impaired. Almost all the phenotypes they observe can be rescued equally well by expression of either Vti1a or Vti1b.

Overall the manuscript is well written, the experiments appear to be well conceived and executed, and they present a great deal of evidence supporting their conclusions. Their results confirm previous studies supporting a role for Vti1a/b in regulating trafficking through the Golgi, and add new information about the redundant and non-redundant functions of the 2 isoforms. They should be of interest to the field.

I have the following suggestions:

1. I find it somewhat surprising that the authors never perform their experiments in wildtype neurons. What is the rationale for only presenting control data from heterozygote knockouts?

2. The methods section implies that DHZ and DKO neurons were plated at different densities. "1500 to 4000 neurons / well for DHZ and DKO, respectively, were plated". Is this correct, and if so, might this affect synapse formation?

3. NPY-phl is used to specifically study release from DCVs. Overexpression often leads to proteins accumulating in organelles where they would not normally be present. Can the authors truly be certain NPY-phl is expressed only in DCVs?

4. A previous study reported different expression patterns for Vti1a and Vti1b in NRK cells (Kreykenbohm et al, EJCB 2002). The authors could discuss the reasons for the observed differences. Also, the antibodies for Vti1b are not specified in the methods.

5. Labeling of timepoints for Fig 7G-H are a bit confusing as they do not agree with the legend.

6. Text change suggestion for Page 12: Cholera Toxin subunit-B (CTB) can be used in neurons as reporter for retrograde transport see, for instance, 38

7. Given CTB retrograde transport relies on both synaptic and neuronal activity, this result could be explained by the reduced number of synapses in DKO neurons, rather than by deficient retrograde transport to the Golgi.

8. The figure colors will be difficult for color blind readers, particularly for discriminating between the DKO vs DKO+Vti1a.
9. The illustrations in Figures 7,8 do not convey information with much clarity. Fig7A could use labels, and Fig8E could be compressed to a single picture that shows their conclusions for the DKO.

REPLY TO THE REVIEWERS, EMPERADOR-MELERO et al, NCOMMS-18-03083

This manuscript has been evaluated by 3 reviewers, who all expressed strong support for publication. We are very grateful for the thorough and supportive evaluation by all reviewers and their valuable comments. Below is a point-by-point reply with the reviewers' text in blue and our responses in black.

REVIEWER 1:

The authors provide here a very complex and very thorough analysis of neurons lacking the SNAREs Vti1a and Vti1b. These molecules have been to some extent analyzed in the neuronal context. For example, Vti1a has been proposed to be involved in the exo- and endocytosis of a spontaneously fusing pool of synaptic vesicles (Ramirez et al., 2012; the authors' reference 26). This is a puzzling observation, as the authors also note in the current manuscript, and it does not fit with the main lines of the synaptic transmission field. It is much more likely that Vti1a and Vti1b, which are presumably found on cellular endosomes, are involved in the biogenesis of synaptic elements, and that their effects on synaptic transmission are not related to a direct involvement in the exo- or endocytosis of synaptic vesicles. The current study provides a much needed perspective in the analysis of SNARE molecules in neurons: the authors study not only the end-point effects on synaptic transmission, but also the influence of these molecules on multiple other processes, including different steps of the secretory pathway. They conclude that Vti1a and Vti1b have largely overlapping functions in the secretory pathway, and that they participate in the biogenesis of synaptic vesicles and of dense-core vesicles, at a step of material recycling in the Golgi apparatus.

I find the current study convincing, well designed, and well executed. As the topic is interested, as noted above, I suggest that the manuscript be published. However, one important point, and a few minor ones, should be explored by the authors before publication."

Major point:

The authors should perform a few simple experiments in non-neuronal cells lacking Vti1a and Vti1b, in order to test whether the effects described here are specific for neurons. Could the authors test, for example, retrograde Cholera Toxin trafficking in cultured fibroblasts derived from their mice? Ideally, the authors could also repeat the assay presented in Figure 7, in fibroblasts. Such experiments will test whether the authors' main conclusion is valid in all cells, or only in neurons. Either conclusion would be important, and would strengthen the current manuscript.

We agree with the reviewer that this is a relevant experiment. We have implemented this suggestion. We studied retrograde trafficking of Cholera toxin and transport of NPY into/out of the Golgi (the assay presented in Fig 7 for neurons) in astrocytes. Astrocytes were used instead of fibroblasts because astrocytes can be cultured using E18 brains, like neurons, whereas fibroblast would require E13.5 embryos. Therefore, astrocytes allowed us to minimize the amount of nests used for this study, in line with Dutch ethical guidelines. Furthermore, fibroblast culture is not a standard assay in our lab. In any case, using astrocytes we were able to address the reviewer's question whether our main conclusions are valid in other cells, not only neurons and we observed that the export of NPY and the retrograde

transport of Cholera toxin were delayed in astrocytes lacking Vti1a and Vti1b, like we observed before in neurons. These data are added to Supplemental Fig 10 to the manuscript and shown as Rebuttal Fig 1, below. We also show that astrocytes express Vti1a and Vti1b in Rebuttal Fig 2, below. We have also included the corresponding text to the results section (p12, l13) and discussion (p17, l1-3), concluding that “Vti1a and Vti1b do not only support this function in neurons”.

Minor points:

1. The authors should perform some conventional immunostaining experiments for native Vti1a and Vti1b, in normal neurons, to indicate their rough cellular localizations. Even better, the colocalization with Golgi and endosomal markers could be analyzed.

We previously observed that Vti1a and Vti1b mainly localize to the Golgi in DHZ neurons (Fig 6A, B). We repeated this in wild-type neurons and also observed preferential localization to the Golgi. We also studied the co-localization of Vti1a and Vti1b with two endo-lysosomal markers (Rab5 and LAMP1) outside of the Golgi and observed partial co-localization, in line with the reported functions of Vti1a/b in early/late endosome and lysosome fusion^{1,2}. We made a dedicated figure for this data (Supplemental Fig 7) and included the corresponding explanation in the results (p10, l5-6) and discussion sections (p14, l3-5 of the section).

2. The authors may want to analyze the significance of the changes shown in Figure 4. As it stands, some of the changes are very likely significant, as for SNAP25, but this is less clear for other proteins, such as Syntaxin 1 or Munc18-1.

The histograms from Fig 4 and the bar plots from Supplemental Fig 5 come from the same data set. The difference is the histograms represent the intensity distribution of all synapses and the bar plots represent the average per neuron. Because synaptic intensity is a nested parameter, we consider that the appropriate place to show statistics is at bar plots, but not at histograms. For this reason, we only added the significant values in Supplemental Fig 5, where we report statistically reduced levels of SNAP25, Munc13-1, Bassoon, Synaptotagmin-1 and RIM1/2 in Vti1a/b DKO neurons, and lower (but not statistically significant) levels of Synaptobrevin-2, Syntaxin-1 and Munc18-1. This is the same rationale that we followed in Fig 1. We have now made clearer in the legend of Fig 4 where to find the significant values.

3. The authors should include in their discussion an additional paper on presumed synaptic effects of Vti1a: “Selective molecular impairment of spontaneous neurotransmission modulates synaptic efficacy”. Crawford et al. Nat Commun. 2017.

We thank the reviewer for this suggestion. We have added this publication to the discussion.

REVIEWER 2:

In this manuscript, Emperador-Melero et al. examined the impact of Vti1a and Vti1b double knockout (DKO) on neuronal survival and morphometric development, trafficking/sorting of neuronal release machinery, and exocytosis of synaptic and dense core vesicles (SVs and DCVs, respectively) using interdisciplinary methodologies including mouse genetics, biochemistry, electron microscopy, electrophysiology and live cell imaging. The authors provide evidence that Vti1a and Vti1b are essential for maintaining neuronal levels of SVs and DCVs and their associated release machineries. Moreover,

they show that SNAP25 and DCV cargo likely accumulate in the secretory pathway at the level of the Golgi apparatus likely due to defective retrograde transport. The paper is generally well-written, and the data will be of interest to the neuroscience community as the essential mediators of protein sorting for maintaining regulated exocytosis in neurons are incompletely understood. Collectively, these data show that Vti1a and Vti1b likely play an important role in maintaining synaptic and other vesicle pools with profound consequences on neuronal development, survival and synaptic function. Nevertheless, while the analyses provided are fairly comprehensive, I would suggest several improvements to strengthen and clarify the conclusions and overall interpretation of the manuscript

Major points:

1. Although relatively comprehensive characterizations were performed and verified by independent and complimentary methodologies, the main message from this paper is somewhat unclear given the generality and widespread nature of the phenotypes (i.e., cell death, reduced axonal and dendritic maturation, impaired vesicle biogenesis, as well as neurotransmitter and dense core vesicle release). We can generally conclude that Vti1a/b are important for neuronal/synaptic function, survival, and maintenance of secretory vesicle pools, but mechanistic conclusions about Vti1a/b and the causal molecular impairments associated with these phenotypes are somewhat lacking. Some clarification and discussion will be very helpful.

We completely agree that the phenotypes reported here are complex and our 'mechanistic conclusions' could have been presented more clearly. We have now revised this in the Discussion section "Impaired retrograde trafficking explains Golgi export defects" (p18-20). This presentation is still necessarily somewhat brief given the space limits of the journal. In short, we conclude by deduction that defects in retrograde delivery of essential components for Golgi export provide a growing problem for vesicle biogenesis, the targeting of synaptic molecules and secretory organelles and eventually synaptic transmission. We also point that all these phenotypes are 'causal', given our experimental design (knock out and rescue).

2. It is unclear whether other regulated secretion is impacted by loss of Vti1a/b. For example, are other synaptic membrane proteins (e.g., cell adhesion proteins, ion channels, g-protein coupled receptors, neurotransmitter receptors, etc.) also reduced in Vti1a/b DKO neurons?

This is certainly an interesting question. We have now added experiments on the TrkB receptor and the postsynaptic protein PSD95. Their levels in neurites were decreased, and we observed accumulation at the Golgi in DKO neurons, similar to SNAP25 and DCV cargo. We made a dedicated figure (Supplemental Fig 9) for this data and included the corresponding explanation to the results section (p11, l8-10). Impaired delivery of calcium and potassium channels has already been demonstrated before^{3,4}. This information appears in the revised manuscript (p17, l6-8 of second pg). Finally, in the original manuscript we already showed that the export of GPI is impaired in DKO neurons. All these arguments are now listed in the discussion (p16-17) to support our conclusion that the export of many different proteins relies on Vti1a/b function.

Are the gross neurodegenerative phenotypes a direct consequence of impaired exocytosis and/or loss or misregulation of other unmeasured proteins that also undergo regulated secretion to the plasma membrane? This will also clarify the specificity of the sorting function by Vti1a/b in neurons.

This is a very difficult question to answer and statements on this topic can only be speculative (especially if we would include 'unmeasured proteins' in our reasoning), because the mechanism of cell death was not the topic of our study. Nevertheless, we agree this is an important issue and we can conclude a few things, also based on previous studies, especially on the question if cell death is "a direct consequence of impaired exocytosis". We have made a dedicated subheading in the discussion entitled "neuronal death in Vti1a/b DKO neurons" (p17, 18) to cover this point. We added the following statement: "Neurons degenerate in the absence of Vti1a/b⁵. Because neuronal death was already abundant at DIV-4, when regulated secretion is minimal⁶, and because previous studies have shown that abolishing regulated secretion per se does not trigger neuronal death⁷, the degeneration observed in DKO neurons is probably not a direct consequence of impaired regulated exocytosis. However, we cannot exclude that the combined loss of membrane proteins (including TrKB receptors) contributes to the degeneration observed."

3. Why is synaptic vesicle number/synapse (EM data, Figure 1K) not impaired since synaptic vesicle biogenesis is also dependent on protein sorting from the Golgi and fluorescence intensity of synaptic markers is markedly decreased in Vti a/b DKO.

We agree it is surprising that the number of SVs is not altered in Vti1a/b neurons and that important implications of this observation were not sufficiently discussed in the original manuscript. In the revised Discussion, we now correct this (p16, first pg): "(...) the number of SVs per synapse was unaffected, while staining intensity for synaptic markers was decreased in DKO synapses. This suggests that, unlike DCV biogenesis, SV biogenesis is unaffected by the loss of Vti1a/b, despite the reduced levels of many synaptic proteins. We conclude that SV biogenesis is coupled to the rate of exocytosis, as suggested⁸, and does not depend on Vti1a/b. Furthermore, given the reduced staining of synaptic proteins, SVs most likely contain a reduced number of native proteins per vesicle in DKO neurons. This provides a plausible explanation for the impaired synaptic transmission in Vti1a/b neurons."

Moreover, DCV number/synapse is reduced by both EM and immunostaining experiments. Do the authors conclude that these vesicles possess fewer copies of protein per vesicle (as suggested by their western-blotting and immunostaining results)? If so, how do they speculate that normal vesicle numbers are maintained? The authors should provide a clear interpretation/discussion on why levels of synaptic vesicle, but not DCV, number is impacted within the overall interpretation of the study.

For DCVs the situation differs from SVs. We conclude that the number of DCVs is reduced in Vti1a/b DKO neurons, but the cargo per vesicle is not changed. Unfortunately, for DCVs we have no quantitative methods to assess the number of most proteins per DCV. We can measure the amount of pHluorin dequenching during fusion as a proxy for the amount of (heterologous) cargo per vesicle. This was unaltered by Vti1a/b loss. We have clarified this now in the discussion (p16, l1-3 of first pg): "Fewer DCVs were observed in synaptic micrographs of DKO neurons and staining for DCV-cargo was reduced, whereas DCV cargo loading per vesicle was unaltered. This indicates that DKO neurons express fewer DCVs with probably unaltered cargo composition."

Minor points:

1. The statement that "Unlike SVs, DCVs do not locally recycle and, hence, the availability of DCVs depends on the delivery on new vesicles, generated at the trans-Golgi" is a bit far reaching. Whether DCV exocytosis exhibits plasticity or and whether DCVs are locally recycled is not well established. We have modified our initial statement (p3, l2-4) to avoid strong statements about DCV recycling.

2. It is unclear why the authors emphasize SNAP25 trafficking and targeting when levels of many other proteins associated with the secretory machinery also appear to be significantly reduced. Simply the fact that SNAP25 levels appear more greatly reduced and they can observe slowed transport of SNAP25 into axons, does not establish that the SNAP25 the major limiting factor in the impairment exocytosis or other phenotypes. The emphasis should be clarified in the discussion and overall interpretation.

SNAP25 was selected because the levels of this protein were reduced the most in DKO neurons. However, we agree that we should avoid the impression that SNAP25 changes explain everything. We have now made this clearer in the Results section (p9, l7-8 of the pg). In the discussion, we have also emphasized that the impaired secretion in DKO neurons can be explained by the combined decrease of many proteins of the secretion machinery, not only SNAP25 (p15, l13-14 of the pg).

3. In the discussion, the comments that "Vti1a is detected in SV fractions and local synaptic functions have been proposed"..."Because SNAP25 supplies the Qb domain in the canonical SNARE complex that drives SV fusion...it is unclear which domain the Qb SNARE Vti1a contributes" and "these indications that Vti proteins support synaptic secretion are generally confirmed in the current study" are, together, a bit confusing. The authors should probably rephrase this section, as it is unlikely they can draw any specific conclusions regarding a direct role for Vti1 proteins in vesicle exocytosis or recycling based on the data shown here.

We agree that the current data do not support strong conclusions on a direct role for Vti proteins in exocytosis. We have rewritten this section to state that point more clearly (p15). We have also included a disclaimer, as the reviewer indicates, stating that our results do not rule out local synaptic functions for Vti proteins.

4. Figure 1B, is synapse formation redundantly regulated by both Vti1a and Vti1b? The number and intensity of synaptophysin puncta appear only partially rescued by re-expression of either Vti1a or Vti1b, whereas DCV (NPY-pHl) number appears fully rescued by both. Can the authors double-check their statistics on these data?

Indeed, the average values for the rescues remained below the control neurons. However, we double-checked the statistics and there was no significant difference between groups and we cannot make any strong claims on the basis of these data. A summary of the statistics is added below this rebuttal.

5. The mean and range for DCV diameters (60nm and 45-120nm, respectively) (Figure 1O&P) are somewhat small for DCVs (typical range 90-250nm, according to literatures). Can the authors double check these quantifications? What are the average sizes of SVs? Note here I am not asking for quantification for SVs, just want to make sure that the SV and DCVs sizes are different and in general agreement of the literature.

The average DCV diameter was 67.1 and 66.2 nm for DHZ and DKO, respectively. This is in line with values previously reported for neuronal DCVs^{9,10}. DCVs are larger (in the range mentioned by the reviewer) in other cell types, such as chromaffin cells³. The average diameter of a SV is ~ 35 nm⁹.

6. What consequences will Vti1a/b DKO have on SV and DCV recycling at their release sites?

We show that the recycling of SVs was not altered in DKO neurons, as indicated by the decay time of the SypHy intensity after high frequency stimulation (Fig 2G, 2E). Whether DCV recycling exists, as this reviewer pointed out earlier, is unclear.

Can the authors show how SV and DCV releasing sites function in the steady-state during their stimulus trains? If the authors claim that "Unlike SVs, DCVs do not locally recycle and, hence, the availability of DCVs depends on the delivery on new vesicles, generated at the trans-Golgi", they should demonstrate differences in the temporal dynamics of release and maintenance of releasing pools during steady-state stimulations in Vti1a/b DKO.

We are not aware of methodology to directly "show how SV and DCV releasing sites function". Of course we can infer, as the reviewer suggests, how releasing sites function from the "temporal dynamics of release and maintenance of releasing pools". However, our main observation was that these dynamics are unaltered. Release is scaled down substantially without significant changes in dynamics (Supplemental Fig 3E and Fig 3F). Hence, our data suggest that there must be less releasing sites. This can be explained by the fact that Vti1a/b are required for normal targeting of release site components such as SNAP25, Munc13-1 or RIM1/2. Similarly, we have shown before that Vti1a KO chromaffin cells contain less secretory granules but also unaltered fusion kinetics, which is explained by a reduction in docked vesicles³. These previous observations align well with our current data in neurons.

7. Related with major point #2, can the authors provide more analyses to show normal ultrastructural development of synapses in Vti1a/b DKO including the number of docked vesicles and PSD width?

We have added the requested data (Supplemental Fig 2K, L) with the corresponding results details (p5, l1-3 of second pg). PSD length and number of docked SVs is unaltered in DKO neurons.

8. Please fix typo in Figure 8 E Vti DKO schematic - "Proteins for synaptic secretion"

We have fixed the typo. Thanks.

REVIEWER #3:

This manuscript reports that Qb-SNAREs Vti1a and Vti1b impact the number of synapses and the levels of synaptic proteins by regulating the sorting and export of proteins from the Golgi. In cultured hippocampal neurons from Vti1a/b double knockout neurons, they find fewer synapses and lower levels of synaptic release machinery proteins. They see fewer dense core vesicles but not synaptic vesicles. Electrophysiological and imaging experiments suggest both synaptic vesicles and dense core vesicles are less likely to be released in response to stimulation. The export of cargo from the Golgi appears to be impaired. Almost all the phenotypes they observe can be rescued equally well by expression of either Vti1a or Vti1b.

Overall the manuscript is well written, the experiments appear to be well conceived and executed, and they present a great deal of evidence supporting their conclusions. Their results confirm previous studies supporting a role for Vti1a/b in regulating trafficking through the Golgi, and add new information

about the redundant and non-redundant functions of the 2 isoforms. They should be of interest to the field.

Suggestions:

1. I find it somewhat surprising that the authors never perform their experiments in wildtype neurons. What is the rationale for only presenting control data from heterozygote knockouts?

The rationale is practical: it is very unlikely to obtain double-KO and double-WT mice in the same nest. To obtain double-KO and double-WT in the same nest would require crossing: Hz/Hz x Hz/Hz with a chance of 1:16 (double-KO *or* double-WT). Hence, in practice nests with both double-KO *and* double-WT hardly ever happen and we would discard most nests. This is not acceptable for Dutch ethics regulations. In contrast, the probability of obtaining a double-Hz and a double-KO in the same litter is 1:4 by crossing: Hz/KO x KO/Hz. Double-Hz are appropriate controls, as indicated by the fact that basic morphologic parameters (neurites length, synapse density, DCV size etc.) and functional parameters (synaptic secretion, DCV trafficking speed, etc.) are similar to previously reported values for WT neurons^{6, 10-12}. We have clarified this in the methods section (p21, l1-2).

2. The methods section implies that DHZ and DKO neurons were plated at different densities. "1500 to 4000 neurons / well for DHZ and DKO, respectively, were plated". Is this correct, and if so, might this affect synapse formation?

This is correct. This difference is to compensate for the increased cell death of DKO neurons. The end goal is to obtain a similar density at the time of the recording. This strategy is also used for other genotypes that also have an increased cell death⁶. We have now clarified this in the methods (p21, l1).

3. NPY-pHl is used to specifically study release from DCVs. Overexpression often leads to proteins accumulating in organelles where they would not normally be present. Can the authors truly be certain NPY-pHl is expressed only in DCVs?

This is a valid remark. For this reporter, it has been validated that targeting is exclusively to DCVs^{9, 13}, but we failed to indicate this clearly in the text. We have now clarified this (p5, l1-2).

4. A previous study reported different expression patterns for Vti1a and Vti1b in NRK cells (Kreykenbohm et al, EJC 2002). The authors could discuss the reasons for the observed differences.

This is also a valid point. The following statement was added to the discussion "Vti1b localized to the Golgi in neurons to a larger extent than what was reported in fibroblast cell lines, which may indicate cell type differences in the dynamics of Golgi and post-Golgi transport" (p18, l9-10 of second pg).

Also, the antibodies for Vti1b are not specified in the methods.

We apologize for this omission. We have added this information to the methods section "Vti1b antibody (1:500; described in¹⁴)".

5. Labeling of timepoints for Fig 7G-H are a bit confusing as they do not agree with the legend.

We thank the reviewer for noticing this mistake. We have amended it.

6. Text change suggestion for Page 12: Cholera Toxin subunit-B (CTB) can be used in neurons as reporter for retrograde transport see, for instance, 38

Thanks, we have corrected it.

7. Given CTB retrograde transport relies on both synaptic and neuronal activity, this result could be explained by the reduced number of synapses in DKO neurons, rather than by deficient retrograde transport to the Golgi.

This is a very good point. To address this, we have now repeated this experiment at DIV-5, when there is no synaptic activity yet. We observed that 2h after incubation with CTB, similar levels were present at the Golgi of DKO and DHZ neurons. However, after a shorter time (30 min), lower levels were observed in DKO neurons, indicating that, indeed, Vti1a/b are required for retrograde transport of CTB to the Golgi and this effect cannot be explained by differences in synapses or activity. We added these data to Supplemental Fig 11, and included the corresponding explanation to the results (p13, l9-11). The difference between DIV-5 and DIV-14 neurons can be explained by “(...) components that regulate this trafficking probably become progressively missorted over time. Alternatively, different Qb-SNARE may be involved in similar pathways during different developmental stages³¹, or synaptic activity influences CTB retrograde trafficking, as suggested³⁸” (and as the reviewer pointed out). This statement was included in the Discussion (p19, l1-4).

8. The figure colors will be difficult for colorblind readers, particularly for discriminating between the DKO vs DKO+Vti1a.

We thank the reviewer for pointing this out. We have changed the green for cyan.

9. The illustrations in Figures 7,8 do not convey information with much clarity. Fig7A could use labels, and Fig8E could be compressed to a single picture that shows their conclusions for the DKOs.

We thank the reviewer for the advice. We have made pertinent changes.

References

1. Antonin, W., *et al.* A SNARE complex mediating fusion of late endosomes defines conserved properties of SNARE structure and function. *The EMBO journal* **19**, 6453-6464 (2000).
2. Pryor, P.R., *et al.* Combinatorial SNARE complexes with VAMP7 or VAMP8 define different late endocytic fusion events. *EMBO reports* **5**, 590-595 (2004).
3. Walter, A.M., *et al.* The SNARE protein vti1a functions in dense-core vesicle biogenesis. *The EMBO journal* **33**, 1681-1697 (2014).
4. Flowerdew, S.E. & Burgoyne, R.D. A VAMP7/Vti1a SNARE complex distinguishes a non-conventional traffic route to the cell surface used by KChIP1 and Kv4 potassium channels. *The Biochemical journal* **418**, 529-540 (2009).
5. Kunwar, A.J., *et al.* Lack of the endosomal SNAREs vti1a and vti1b led to significant impairments in neuronal development. *Proceedings of the National Academy of Sciences of the United States of America* **108**, 2575-2580 (2011).
6. Arora, S., *et al.* SNAP-25 gene family members differentially support secretory vesicle fusion. *Journal of cell science* **130**, 1877-1889 (2017).
7. Santos, T.C., Wierda, K., Broeke, J.H., Toonen, R.F. & Verhage, M. Early Golgi Abnormalities and Neurodegeneration upon Loss of Presynaptic Proteins Munc18-1, Syntaxin-1, or SNAP-25. *The Journal of neuroscience : the official journal of the Society for Neuroscience* **37**, 4525-4539 (2017).
8. Haucke, V., Neher, E. & Sigrist, S.J. Protein scaffolds in the coupling of synaptic exocytosis and endocytosis. *Nature reviews. Neuroscience* **12**, 127-138 (2011).
9. Emperador Melero, J., *et al.* Differential Maturation of the Two Regulated Secretory Pathways in Human iPSC-Derived Neurons. *Stem cell reports* **8**, 659-672 (2017).

10. van de Bospoort, R., *et al.* Munc13 controls the location and efficiency of dense-core vesicle release in neurons. *The Journal of cell biology* **199**, 883-891 (2012).
11. Gummy, L.F., *et al.* MAP2 Defines a Pre-axonal Filtering Zone to Regulate KIF1- versus KIF5-Dependent Cargo Transport in Sensory Neurons. *Neuron* **94**, 347-362.e347 (2017).
12. de Wit, J., Toonen, R.F., Verhaagen, J. & Verhage, M. Vesicular trafficking of semaphorin 3A is activity-dependent and differs between axons and dendrites. *Traffic (Copenhagen, Denmark)* **7**, 1060-1077 (2006).
13. Farina, M., *et al.* CAPS-1 promotes fusion competence of stationary dense-core vesicles in presynaptic terminals of mammalian neurons. *eLife* **4** (2015).
14. Antonin, W., Riedel, D. & von Mollard, G.F. The SNARE Vti1a-beta is localized to small synaptic vesicles and participates in a novel SNARE complex. *The Journal of neuroscience : the official journal of the Society for Neuroscience* **20**, 5724-5732 (2000).

Rebuttal Figures

Appendix 1: Summary statistics Fig 1B

Synapse density:

DHZ: 0.24 ± 0.01 ; DKO: 0.15 ± 0.01 ; DKO+Vti1a: 0.20 ± 0.01 ; DKO+Vti1b: 0.22 ± 0.01 synapses per μm dendrite. Data was not normally distributed in the DKO group ($W = 0.90476$, $p\text{-value} = 0.008202$). Then, a non-parametric test was used (Kruskal-Wallis chi-squared = 22.608, $df = 3$, $p\text{-value} = 4.875e-05$). Post Hoc analysis:

	p-value
DHZ vs DKO	3.7e-05***
DHZ vs DKO+1a	0.051
DHZ vs DKO+1b	0.219
DKO vs DKO+1a	0.011*
DKO vs DKO+1b	0.018*
DKO+1a vs DKO+1b	0.464

Synapse intensity:

DHZ: 151.71 ± 5.82 ; DKO: 106.96 ± 5.16 ; DKO+Vti1a: 133.47 ± 5.24 ; DKO+Vti1b: 137.75 ± 8.09 AU. All data was normally distributed and variances were homogeneous. Then, an ANOVA was used: ANOVA (3) = 7.585 ; $p = 8.42e-05$ ***. Post Hoc analysis:

	p-value
DHZ vs DKO	0.0000258***
DHZ vs DKO+1a	0.1064122
DHZ vs DKO+1b	0.3940868
DKO vs DKO+1a	0.0207474*
DKO vs DKO+1b	0.0110830*
DKO+1a vs DKO+1b	0.9578766

REVIEWERS' COMMENTS:

Reviewer #1 (Remarks to the Author):

The authors replied to all of my comments thoroughly, and I am happy to suggest that the manuscript be published in the current form.

Reviewer #2 (Remarks to the Author):

The authors had addressed my previous concerns in this revised version. I support the publication of the paper in Nature Communications.

Reviewer #3 (Remarks to the Author):

The authors have done a thorough job of revising the manuscript, adding multiple new experiments and supplemental figures to address the reviewers' concerns. They also amend the language of their conclusions to more accurately reflect their data. The result is a comprehensive - if complicated- examination of the role of Vti1a/b in trafficking to the Golgi. While their results raise many questions, they will be of interest to the community and should be published.

The new data showing CTB retrograde transport in DIV-5 neurons (Supplemental Fig 11) is somewhat confusing. 2 hours after CTB incubation they saw no difference between DKO and control neurons, so the authors repeated the experiment with a shorter incubation time. The authors do not explain their rationale for this shorter incubation time, or the rationale for studying DIV-5 neurons (lack of synaptic transmission is stated in the response to reviewers only). Please add an explanation to the results section. Also, given the time-dependence of CTB transport they see at DIV-5 (and the authors' conclusions about the progressive sorting deficits), it would have been prudent to perform similar 30 minute experiments at DIV-14.

Minor points:

Figure 5E,F: The green in DKO+Vti1a graphs should be switched to cyan to match other figures.

Supplemental Fig 3L: The scale bar for time appears to be incorrect. It does not match a 40 Hz stimulation.

Supplemental Fig 4B,C: The authors are not measuring calcium influx, but rather Fluo5F fluorescence.

REPLY TO THE REVIEWERS, EMPERADOR-MELERO et al, NCOMMS-18-03083A

REVIEWER 1:

The authors replied to all of my comments thoroughly, and I am happy to suggest that the manuscript be published in the current form.

We thank the reviewer for the positive evaluation.

REVIEWER 2:

The authors had addressed my previous concerns in this revised version. I support the publication of the paper in Nature Communications.

We thank the reviewer for the positive evaluation.

REVIEWER #3:

The authors have done a thorough job of revising the manuscript, adding multiple new experiments and supplemental figures to address the reviewers' concerns. They also amend the language of their conclusions to more accurately reflect their data. The result is a comprehensive -if complicated- examination of the role of Vti1a/b in trafficking to the Golgi. While their results raise many questions, they will be of interest to the community and should be published.

The new data showing CTB retrograde transport in DIV-5 neurons (Supplemental Fig 11) is somewhat confusing. 2 hours after CTB incubation they saw no difference between DKO and control neurons, so the authors repeated the experiment with a shorter incubation time. The authors do not explain their rationale for this shorter incubation time, or the rationale for studying DIV-5 neurons (lack of synaptic transmission is stated in the response to reviewers only). Please add an explanation to the results section. Also, given the time-dependence of CTB transport they see at DIV-5 (and the authors' conclusions about the progressive sorting deficits), it would have been prudent to perform similar 30 minute experiments at DIV-14.

We thank the reviewer for the positive evaluation. As requested by the reviewer, we have rewritten the corresponding results section to clarify our rationale to use different DIVs and time points after incubation (p12). This is our explanation:

(...) Because we previously observed that defects in GPI export from the Golgi were only detected after short, but not long, incubation times (Figure 7L-M), we reasoned that a similar time-dependency may affect CTB-A488 and explain the absence of a measurable phenotype 2h after CTB-A488 incubation in DIV-5 neurons. To explore this possibility, we measured CTB-A488 expression at the Golgi 30min after incubation and observed that, at this (shorter) time point, the relative fluorescence at the Golgi was decreased in DKO neuron (Supplemental Figure 11A-D).

Minor points:

Figure 5E,F: The green in DKO+Vti1a graphs should be switched to cyan to match other figures.

- Supplemental Fig 3L: The scale bar for time appears to be incorrect. It does not match a 40 Hz stimulation.
- Supplemental Fig 4B,C: The authors are not measuring calcium influx, but rather Fluo5F fluorescence.

We thank the reviewer for the observations. We have amended these points.

1. Wang, T. et al. Flux of signalling endosomes undergoing axonal retrograde transport is encoded by presynaptic activity and TrkB. *Nat Commun* **7**, 12976 (2016).